# Marine biogenic humic substances control iron biogeochemistry across the Southern Ocean

C. S. Hassler [1,2,3] ✉, R. Simó [4], S. E. Fawcett [5,6], M. J. Ellwood [7,8] & S. L. Jaccard [3]

Iron, which is an essential element for marine photosynthesis, is sparingly soluble in seawater. In consequence, iron bioavailability controls primary productivity in up to 40% of the world's ocean, including most of the Southern Ocean. Organic ligands are critical to maintaining iron in solution, but their nature is largely unknown. Here, we use a comprehensive dataset of electro-active humics and iron-binding ligands in contrasting regions across the Southern Ocean to show that humic substances are an important part of the iron binding ligand pool, as has been found elsewhere. However, we demonstrate that humics are mostly produced in situ and composed of exopolymeric substances from phytoplankton and bacteria, in contrast to other regions where terrestrially-derived humics are suggested to play a major role. While phytoplankton humics control the biogeochemistry, bioavailability and cycling of iron in surface waters, humics produced or reprocessed by bacteria affect iron cycling and residence time at the scale of the global ocean. Our findings indicate that autochthonous, freshly released organic matter plays a critical role in controlling primary productivity and ocean-climate feedbacks in iron-limited oceanic regions.

The ocean modulates both the global carbon cycle and Earth's climate. Each year, the ocean absorbs roughly one-fourth of anthropogenic $CO_2$ emissions[1,2], de facto mitigating a large proportion of their global warming potential. Around 40% of this carbon uptake occurs in the Southern Ocean[3] (SO), where cold waters absorb atmospheric $CO_2$ and transfer it into the ocean interior by convection and mixing. This solubility pump is enhanced by the biological carbon pump, whereby marine phytoplankton fix $CO_2$ into organic carbon biomass, reducing $CO_2$ concentrations in surface waters and amplifying ocean carbon uptake[4]. Phytoplankton carbon is partly recycled in shallow waters and partly exported to depths where it will remain sequestered for centuries to millennia[2,4]. Iron (Fe), essential for photosynthesis[5], is the dominant micronutrient limiting phytoplankton growth over most of the SO, especially south of the Polar Front[6–9]. Thus, the scarcity of bioavailable Fe limits the efficiency of the biological carbon pump, thereby curtailing atmospheric $CO_2$ drawdown. The contribution of the SO to global carbon uptake is expected to increase with global warming, mostly due to increased Fe and light availability[10].

While it is widely accepted that Fe chemical forms largely explain Fe bioavailability[9,11], Fe chemistry alone cannot predict whether this micronutrient will sustain phytoplankton growth[9,12,13]. In marine ecosystems, more than 99% of the dissolved Fe is bound to organic ligands

[1]Department F.-A. Forel for Environmental and Aquatic Sciences, University of Geneva, Geneva, Switzerland. [2]School of Architecture, Civil, and Environmental Engineering, Smart Environmental Sensing in Extreme Environements, ALPOLE, Ecole Polytechnique Fédérale de Lausanne, Sion, Switzerland. [3]Institute of Earth Sciences, University of Lausanne, Lausanne, Switzerland. [4]Institut de Ciències del Mar, ICM-CSIC, Barcelona, Catalonia, Spain. [5]Department of Oceanography, University of Cape Town, Rondebosch, South Africa. [6]Marine and Antarctic Research Centre for Innovation and Sustainability, University of Cape Town, Rondebosch, South Africa. [7]Research School of Earth Sciences, Australian National University, Canberra, Australia. [8]Australian Centre for Excellence in Antarctic Science (ACEAS), Australian National University, Canberra, Australia. ✉e-mail: Christel.Hassler@epfl.ch

within the dissolved organic matter (DOM) pool[6], one of Earth's largest organic carbon reservoirs that nonetheless remains poorly characterised. Iron-binding organic ligands (Fe-L) exert a major control on (i) the oceanic residence time of dissolved Fe, (ii) the subsurface ocean Fe inventory, and (iii) Fe bioavailability, i.e., the acquisition of Fe-substrate by microorganisms[6,9,12–14]. Iron bioavailability is not controlled by seawater chemistry only, but it is also influenced by the biological uptake mechanisms at play, as well as biological competition for Fe acquisition[9,11–13]. Iron bioavailability varies far more widely in the SO (>200-fold variation)[13] than in the global ocean (5-fold variation)[12], highlighting that our poor knowledge of the nature of Fe-L is a major impediment to our understanding of the biological carbon pump in the SO.

It has recently been proposed that Fe biogeochemistry across the global ocean is regulated by humic substances (HS). This conclusion was based on measurements of fluorescence and chemical DOM properties[14–19], as well as through coupling of Fe-binding stoichiometry from terrestrially-derived standard humics (Suwanee River humics) to electrochemically-detected iron-bound HS (eHS)[14,19–23]. HS represent a continuum of organic compounds resulting from the degradation of (terrestrial or aquatic) biomass, including the degradation of polysaccharides, proteins, lipids, nucleic acids, and lignin[14,17]. Within HS, humic and fulvic acids are two operationally-defined fractions, with humic acids (HA) being soluble in natural and alkaline solutions only, and fulvic acids (FA) being soluble at all pH levels[17].

The important role of terrestrial humics in the HS pool has been clearly demonstrated in the transpolar drift of the Arctic Ocean[20,21] and in coastal systems[18], as well as through their long-term persistence in marine systems[17,18]. However, most studies have also recognised the importance of other sources of humics including atmospheric dust deposition[24], microbial reworking of marine DOM[25–27] and remineralisation in the ocean's water column[28]. The widespread abundance of marine humics[14,17] has even been determined in the landlocked Mediterranean Sea[22,29] and in coastal systems such as at the East Antarctic Peninsula[30]. Humics have also been reported in SO[14,23], yet their sources and role in Fe biogeochemistry remain largely unexplored despite recent data suggesting that Fe-binding ligands are critical drivers of Fe

residence time and bioavailability in surface waters[31]. Our study addresses this important knowledge gap.

## Results

### Uniqueness of iron-binding organic ligands across the Southern Ocean

During the Antarctic Circumnavigation Expedition (ACE) in the summer 2016–2017, we conducted a circumpolar assessment of the connections among ice, ocean, climate, and life across contrasting regions of the SO (Fig. 1, Supplementary Figs. 1 and 2, Supplementary Table 1)[13]. Unprecedented quasi-synoptic measurements ($n = 70$) of eHS, hydrolysable carbohydrates (Carb), dissolved Fe (DFe), and Fe-L concentrations were carried out on seawater samples collected all around Antarctica along vertical profiles spanning the upper 1000 m of the water column, to assess the role of eHS in Fe biogeochemistry. Carb showed an inverse correlation with depth (Fig. 1; $R = -0.21$, $p < 0.001$, $n = 239$, Supplementary Table 2), suggesting vertical attenuation of carbohydrates and CHONS-containing molecules as a result of microbial DOM processing[24–27,32]. A strong positive correlation between Fe-L and eHS (Fig. 1b; $R = 0.7$, $n = 70$, $p = 1.8 \times 10^{-11}$, Supplementary Table 2, Fig. 2d) suggests that eHS, despite being a subset of the HS pool, might represent a substantial fraction of the Fe-L across the SO. Notably, the correlation also holds ($R = 0.8$, $n = 39$, $p = 9.2 \times 10^{-10}$, Supplementary Table 3) when considering surface waters (0–100 m) only. Overall, the intercept was 1.3 nmol L$^{-1}$ and the slope was 0.045 nmol μg$^{-1}$ eHS (Supplementary Fig. 4). Thus, on average, 1 mg eHS could be associated with 45 nmol L$^{-1}$ Fe-L, which is close to the terrestrial SRHA value (32 nmol Fe mg$^{-1}$ SRHA)[14,33,34]. Fe-L and eHS values associated with sedimentary input close to the Balleny Islands[13,35] were amongst the highest measured and could explain a Fe-binding capacity for eHS close to that of terrestrial humics.

Although this finding supports the postulated role of eHS in Fe biogeochemistry across the global ocean[14,19–23,36], no correlation between DFe and eHS was observed for the SO (Fig. 2b) in contrast to all other oceanic regions for which comparable data are available (Fig. 2a, Supplementary Fig. 3, except for the South Pacific Ocean). In the global ocean, typically >99% of the DFe is associated with Fe-L[6]; therefore, if

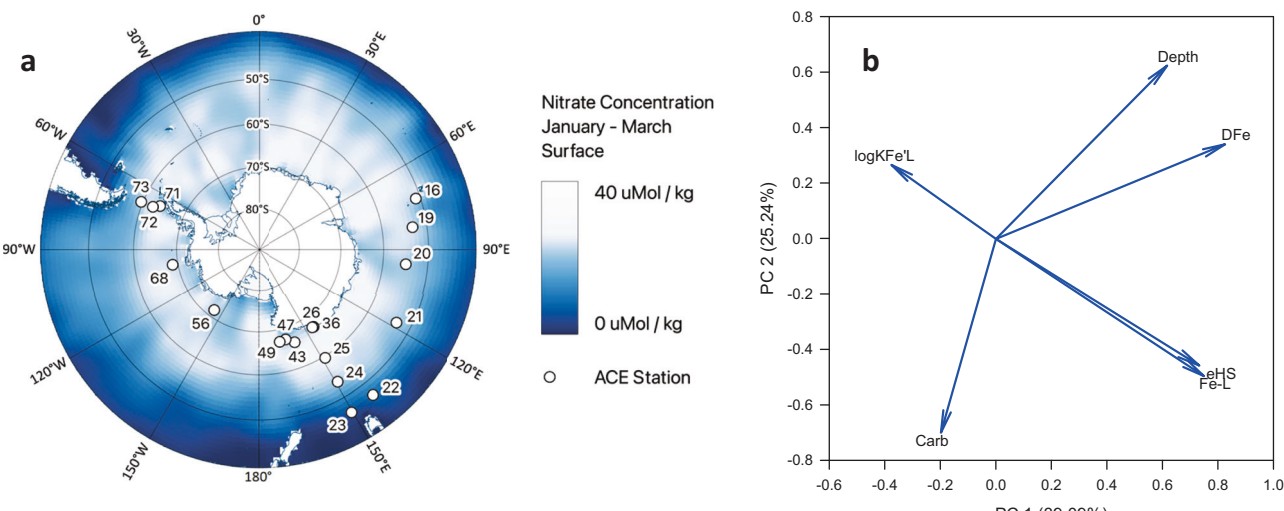

**Fig. 1 | Key relationships across the Southern Ocean. a** Sampling stations are shown superimposed on nitrate concentrations to illustrate the contrasting sites sampled during the Antarctic Circumpolar Expedition (ACE). **b** Component loading from the Principal Component Analysis for the ACE dataset (0–1000 m, $n = 70$). This figure represents the relationship between dissolved Fe (DFe, nmol L$^{-1}$), in situ iron-binding ligands (Fe-L, nmol L$^{-1}$), the conditional stability constant (log K$_{Fe'L}$),

electroactive humic substances (eHS, μg L$^{-1}$ Suwanee River Fulvic Acid (SRFA) equivalent), hydrolysable carbohydrates (Carb, μg L$^{-1}$ glucose equivalent) and depth (m). Principal Component (PC) 1 and 2 together explain 64.33% of the dataset. The variances of the data are explained at 77% for depth, 80% for DFe, 81% for Fe-L, 22% for log$_{KFe'L}$, 53% for Carb, and 75% for eHS. log K$_{Fe'L}$ and Carb are not statistically represented.

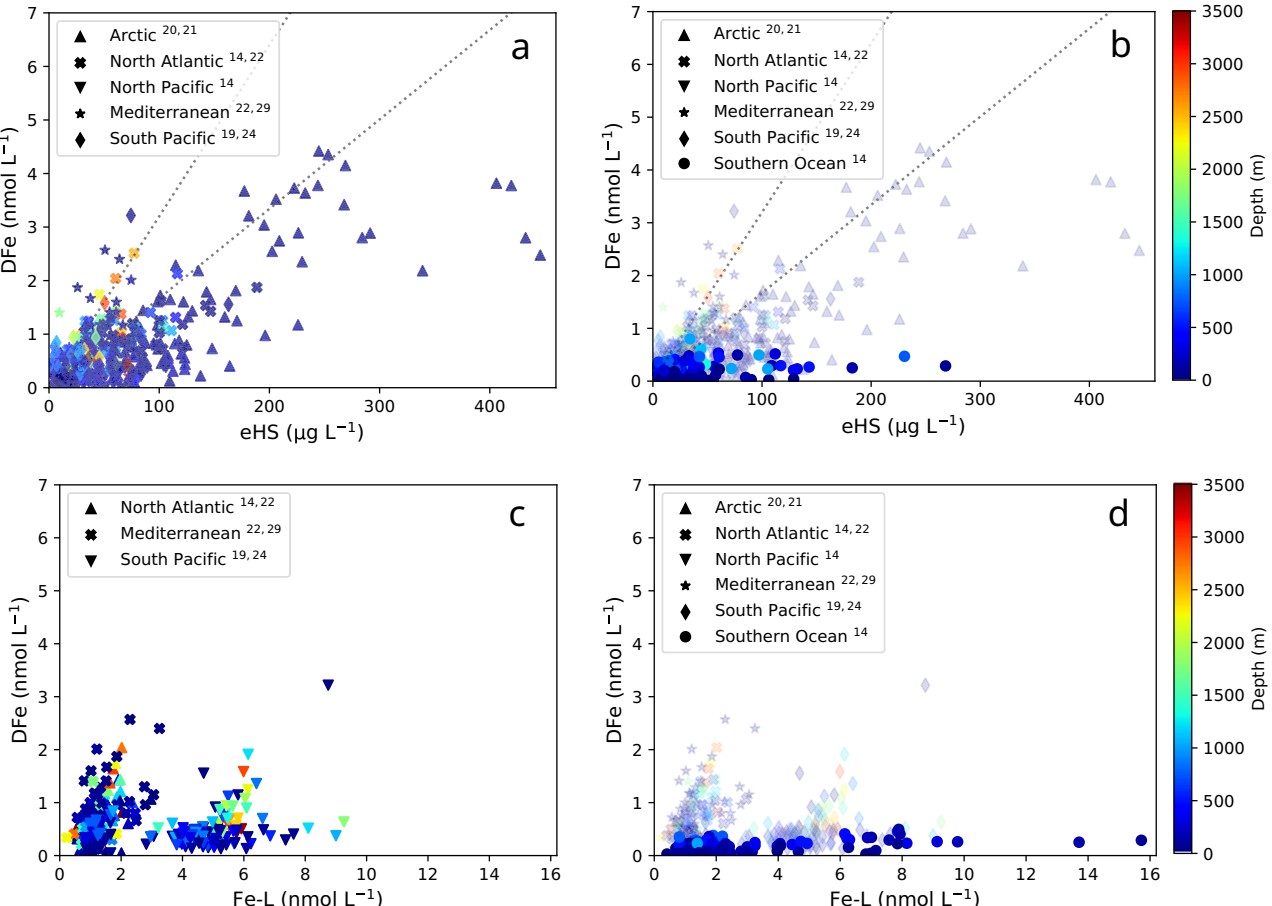

**Fig. 2 | Relationship between dissolved iron, humics, iron-binding ligands, and depth in the global ocean.** Representation of dissolved Fe (DFe) as a function of electroactive humics substances (eHS) in (**a**) different ocean basins and the Mediterranean Sea and (**b**) in the Southern Ocean. Calculated DFe saturation of eHS is shown (dotted lines) considering Suwanee river humic (HA, top) and fulvic acids (FA, bottom). Relationship between DFe and iron-binding ligands (Fe-L) is shown for (**c**) different ocean basins and the Mediterranean Sea, as well as for (**d**) the Southern Ocean. Data are colour-coded as a function of depth. The different ocean basins considered are the Arctic (triangle up[20,21]), North Atlantic (cross[14,22]), North Pacific (triangle down[14]), South Pacific (diamond[19,24]), Southern Ocean (circle[14], *this study*) as well as the Mediterranean Sea (star[22,29]). All the Fe-L data were obtained using salicylaldoxime (SA) as the exchange ligands, except for in the Mediterranean Sea where 2-(2-thiazolylazo)-*p*-cresol (TAC) was used. For the Southern Ocean, our study contributes to 106 out of the 118 observations on (**b**), and all the data represented in (**d**) as no Fe-L data are available in ref. 14.

eHS represents the bulk of Fe-L, one would expect a correlation between DFe and eHS unless (i) eHS are undersaturated with Fe, as might be anticipated in an Fe-limited region such as the SO, and/or (ii) a strong in situ competition between different metals and cations to bind the same eHS molecule exists[6,13,14]. Similar eHS concentrations have been reported for several ocean basins and the Mediterranean Sea (Fig. 3a). The Arctic Ocean emerges as a notable exception, where eHS concentrations are elevated, likely due to strong riverine inputs and high concentrations of terrestrial humics in the transpolar drift[20,21]. The DFe levels associated with eHS are within or smaller than the range observed for saturated Suwannee River FA and HA for a significant portion of the data collected in the North Pacific, North Atlantic and South Pacific Oceans. In the other regions, even lower values were measured (Fig. 3b). Together, competition between other metals and cations to bind eHS, and low DFe resulting in undersaturated eHS will typically lower the measured DFe:eHS values. Surprisingly, similar average DFe:eHS were found across regions globally ($18.4 \pm 19.1$ nmol Fe mg$^{-1}$ eHS, Fig. 3b); a value that is representative of SRFA and SRHA maximal Fe-binding capacity or stoichiometry[14,33]. This finding suggests that the role of humics in the global ocean could be assessed using Suwanee River reference material, although this approach might not be suitable for investigating the southernmost waters. For the SO, a median DFe:eHS of 8 nmol Fe mg$^{-1}$

eHS was observed, suggesting an undersaturation of eHS and/or Fe-L properties or sources different from SRHA and SRFA. Indeed, the relationships between DFe, eHS, and Fe-L indicate a distinct nature of organic matter – Fe interactions in this southernmost ocean (Fig. 2, Supplementary Figs. 3 and 5).

To explore whether, by considering the iron-binding stochiometry from standard terrestrial HA and FA, humics represent the bulk of the Fe-L and could dominate Fe biogeochemistry, we compiled measurements of DFe, in situ Fe-L and eHS from several oceanic regions[14,19–22,24] and estimated the humic contribution (as $L_{FA}$; Fig. 3c, and $L_{HA}$; Fig. 3d) to in situ iron-binding ligands, as in refs. 19,21,22. We note that this approach inherently assumes that all DFe is bound to eHS with similar Fe binding capacity as for standard humics in a 1:1 Fe:eHS stoichiometry, which is a necessary simplification given the complex chemistry of aqueous DOM and Fe[20,37], the fact that Fe can bind to other ligands on a "first come, first served" basis[38], and that other trace elements can compete for the available binding sites. Therefore, the Fe-L$_{eHS}$ attributed to FA and HA should represent an upper estimate of in situ Fe-binding ligands. Only a strong Fe-binding capacity (as for HA) succeeded in representing the bulk of the in situ ligands in the Arctic, North Atlantic and Mediterranean Sea (Fig. 3c, d). In the Arctic, previous direct measurements suggested that $80 \pm 51\%$ of

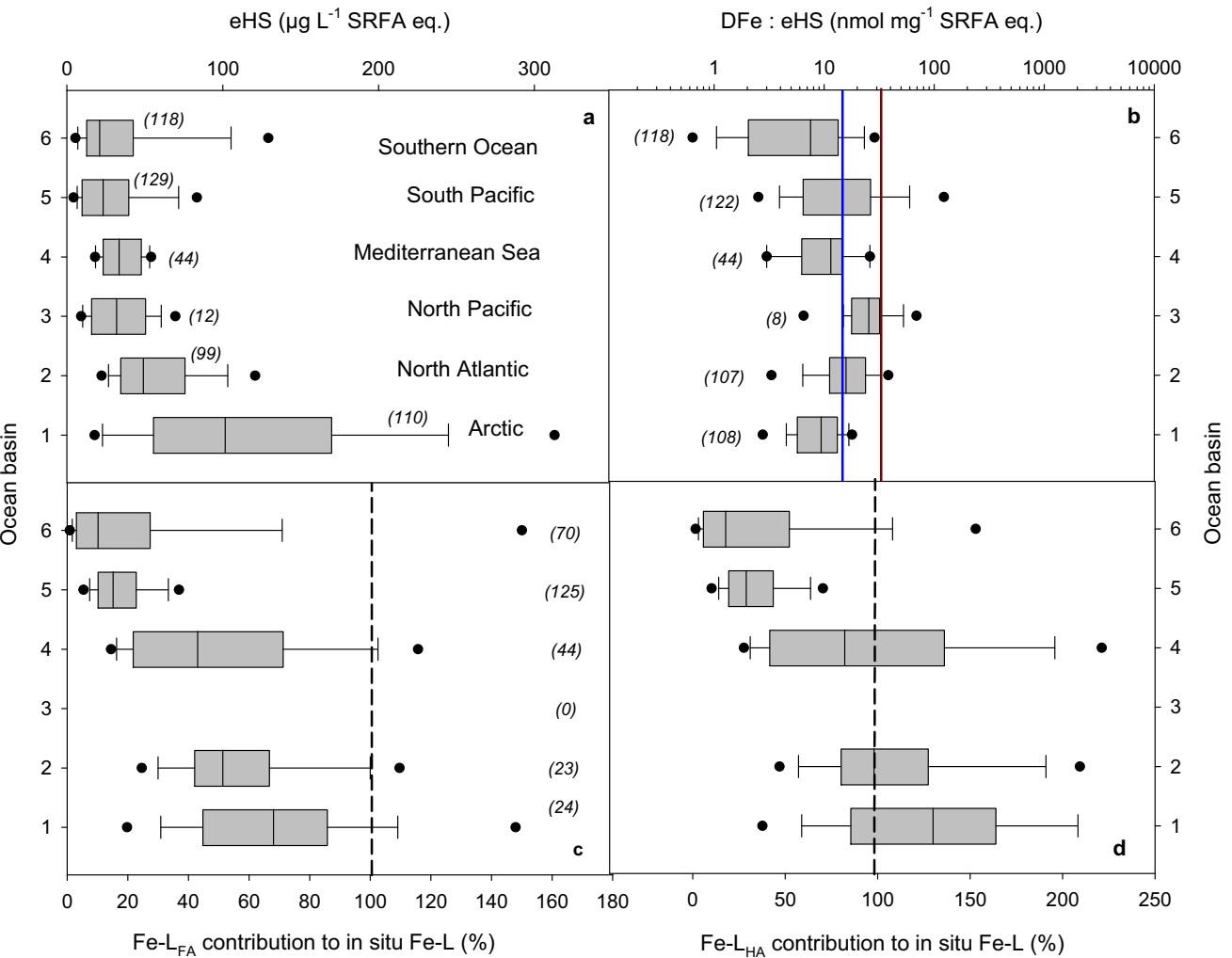

**Fig. 3 | Revisiting the contribution of humic substances to in situ iron-binding ligands. a** Electroactive humic substances (eHS, **a**) and **b** their potential iron saturation level (DFe:eHS ratio) as reported for different aquatic systems. Iron saturation levels for Suwanee River fulvic acid (FA, blue line) and humic acid (HA, brown line) are shown on (**b**). For aquatic systems where both eHS and iron-binding ligand (Fe-L) were available, the relative contribution of eHS to the Fe-L pool was calculated using either Suwanee River FA (**c**) or HA (**d**) iron-binding capacity, as in refs. 14,21. The 100% contribution of humics to in situ Fe-L is shown with a dotted line. Data are represented as box plots with mean and 5th and 95th percentiles. Standard deviations are shown as error bars and the number of observations is listed in brackets next to the corresponding marine region: 1, Arctic[20,21]; 2, North Atlantic[14,22]; 3, North Pacific[14]; 4, Mediterranean Sea[22,29]; 5, South Pacific[19,24]; and 6, Southern Ocean[14] (mostly this study), $n = 106$ for (**a** and **b**), $n = 70$ for (**c** and **d**).

Fe was bound to eHS[20], which falls within the range of $L_{FA}$ and $L_{HA}$ contributions estimated here. In the South Pacific and SO, such an approach fails to represent the bulk of in situ ligands, except for marginal values in the SO associated with sediment inputs near the Balleny Islands[13,35]. Overall, estimated Fe-L$_{eHS}$ only accounted for 10–30% of in situ Fe-L, suggesting that the bulk of eHS measured within the upper 1000 m of the SO and South Pacific cannot be described by standard FA and HA substances. This observation, coupled with the absence of a relationship between DFe and eHS, highlights a different behaviour and/or nature of eHS in these southernmost ocean regions.

### The origin of Fe organic ligands in the Southern Ocean and their relevance to Fe biogeochemistry

Coexisting nutrients and organic matter sources and degradation processes often weaken the observed relationships between the biomass of phytoplankton and bacteria with Fe-L compared to what is expected based on our understanding of known processes at play[30], yet several studies have demonstrated the role of microorganisms in Fe-L production and transformation[28,39–41]. Here, we take advantage of our unique dataset to explore Carb, Fe-L, and eHS relationships with phytoplankton, bacteria, and depth as a way to diagnose key sources and sinks.

Fe-L and eHS concentrations were negatively correlated with bacterial abundances ($R = -0.4$, $n = 45$, $p = 0.0045$; and $R = -0.4$, $n = 46$, $p = 0.0064$, respectively) (Supplementary Table 2). These relationships suggest that microbial reworking of DOM during remineralisation affects both Fe-L and eHS, as previously observed[28,39]. This idea is further supported by the observation that the negative correlation was not observed when only surface waters were considered (0–100 m; $n = 32$, $p = 0.38$, Supplementary Table 3). In general, bacterial and phytoplankton abundances were positively correlated with Carb (Supplementary Table 3), suggesting that these compounds were predominantly photosynthates and exudates. While only weak correlations between phytoplankton and DFe were observed for the ACE dataset, the efficiency of photosystem II, indicated by the Fv/Fm ratio[5], was positively related to both Fe-L ($R = 0.5$, $n = 45$, $p = 0.0003$) and eHS ($R = 0.3$, $n = 47$, $p = 0.036$). Despite the rather low R values, these correlations suggest a ligand-mediated Fe bioavailability to phytoplankton that cannot be predicted from DFe alone, as previously

**Table 1 | Chemical characterisation of different humic materials**

| Substances | mg C-Carb mg⁻¹ S | mg eHS mg⁻¹ S | nM Fe-L mg⁻¹ S | Log K_{Fe'L} |
|---|---|---|---|---|
| **EPS bacteria** | 0.189 ± 0.034 (3)[9,39] | 0.011 ± 0.008 (2)[9,39] | 4.724 ± 2.374 (4)[9,39,***] | 11.38 ± 0.25 (4)[9,39,***] |
| **EPS phytoplankton** | 0.141 ± 0.040 (3)[9,39] | 0.104 ± 0.025 (2)[39] | 62.63 ± 31.22 (2)[39] | 11.65 ± 0.27 (2)[39] |
| **EPS in situ** | 0.109 ± 0.011 (2)[9,39] | 0.164 (1)[39] | 68.07 (1)[39] | 10.91 (1)[39] |
| **Carb - PS** | 0.101 (1)[9] | 0.034 (1)[9] | 0.376 (1)[9] | 10.90 (1)[9] |
| **Carb -MS** | 1.010 (1)[9] | 0.075 (1)[9] | 1.231 (1)[9] | 9.855 ± 1.06 (2)[9,46] |
| **FA** | 0.077 (1)[*,**] | 1.02 (1)[9] | 16.70 (1)[32] | 10.60 (1)[32] |
| **HA** | 0.060 (1)[*,**] | -- | 31.90 (1)[32] | 11.10 (1)[32] |

Average contribution to hydrolysable carbohydrates (Carb, expressed as glucose equivalent), electroactive humic substances (eHS, expressed as Suwanee River fulvic acids equivalent), iron-binding ligands (Fe-L) and their conditional stability constant (log $K_{Fe'L}$) for different organic substances (S). Organic substances were chosen as compounds potentially controlling Fe biogeochemistry and include: Bacterial, phytoplankton and in situ exopolymeric substances (EPS), polysaccharides (carrageenan, PS), monosaccharides (glucuronic acid, MS) and Suwanee River humic substances (fulvic acids, FA, and humic acids, HA). Number of observations (n) is shown in brackets; average is presented with standard deviation for $n \geq 3$ and with half-data gaps for $n = 2$, ND not detected.
*estimated based on carboxylic carbon content** , International humic substances data for SRFA and SRHA std 1*** , Hassler unpublished data.

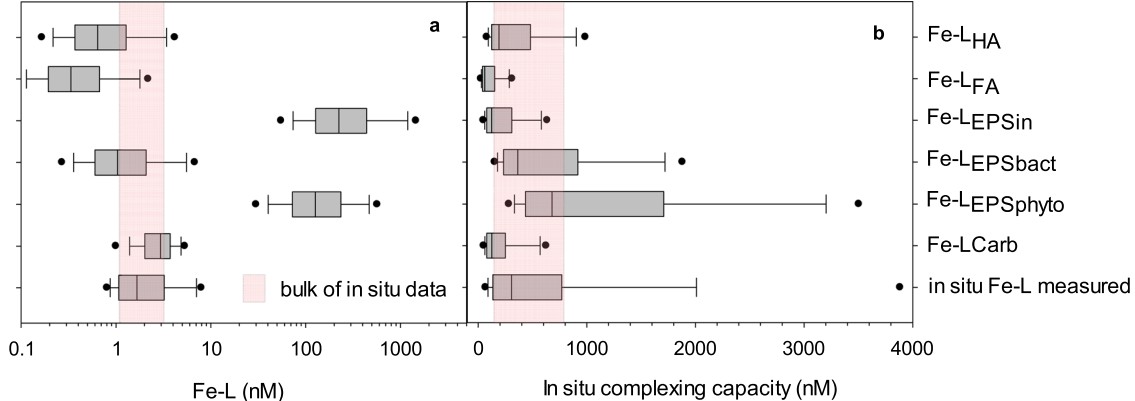

**Fig. 4 | Potential role of different humic materials and carbohydrates in the in situ Fe biogeochemistry. a** Iron-binding ligands (Fe-L) calculated from electroactive humic substances (eHS) assuming that eHS behave as Suwannee river humics (HA), fulvics (FA), exopolymeric substances (EPS) either as in situ EPS (EPS_in), EPS from bacteria (EPS_bact), EPS from phytoplankton (EPS_phyto) or hydrolysable carbohydrates (Carb). **b** In situ complexing capacity for these different humics is used to assess their potential role in Fe biogeochemistry. The complexing capacity was calculated using the measured excess ligands and the iron- binding-conditional stability constants for HA, FA, EPS and Carb (see "Methods" section). Data were compared to the measured in situ Fe-L (**a**) and in situ complexing capacity (**b**). The red regions coincide with the bulk of the in situ observations and aid in visualising the potential role of the different humics considered, with humics ≥ than in situ Fe-L potentially strongly contributing to Fe-L and controlling Fe biogeochemistry. Data are represented as box plots with mean and 5th and 95th percentiles. Standard deviations are shown as error bars.

observed for natural phytoplankton assemblages[9]. The discrepancy between DFe concentrations and ligand-bound Fe bioavailability could be related to the occurrence of a significant proportion of DFe that is less bioavailable because it is associated with colloids. Colloidal Fe plays a critical role in Fe residence time, with consequences for the dynamics of marine ecosystems[31]; unfortunately, we have no measurements of colloidal Fe and thus cannot explore whether these forms are responsible for the lack of correlation between DFe and the biomass of phytoplankton or bacteria.

We identified both allochthonous and in situ-produced compounds as potentially important for Fe biogeochemistry across the SO; unfortunately, our measurements of only eHS and Fe-L prevented us from differentiating them. Among in situ biologically produced Fe-binding ligands, exopolymeric substances (EPS) are described as key compounds[9,12,40–42]. Similar to HS, EPS are loosely defined organic macromolecules able to bind several metals and cations[41,42]. Heterotrophic bacteria and phytoplankton excrete EPS[41–43], which contribute to the eHS pool (Table 1) and are known to represent an important fraction of labile DOM[25,44]. EPS act as weak Fe- binding organic ligands[41,43] (Table 1) that make Fe bioavailable to phytoplankton[12,40,41]. Carbohydrates are often considered key constitutive components given that EPS are generally rich in high molecular weight acidic polysaccharides, many of which contain carboxylic groups[45] - a good functional site for Fe binding. For these

reasons, we further investigated the role of EPS and carbohydrates in SO Fe biogeochemistry.

High-resolution depth profiles of EPS in the ocean do not exist, and the distinction between phytoplankton and bacterial EPS has never been made. Here, we use the chemical characterisation of EPS previously isolated from phytoplankton, bacteria, or a naturally occurring bloom (see "Methods" section) to explore their potential role in Fe biogeochemistry. EPS excreted by phytoplankton and bacteria had similar labile Carb (similar to polysaccharides) but contrasting eHS content (Table 1), suggesting distinct compositions yet similar organic carbon lability. Bacterial EPS had a lower eHS and Fe-L content than phytoplankton EPS, but a similar conditional stability constant with Fe. Together, these observations indicate that the ability of EPS to bind Fe cannot be predicted from their Carb content.

We compiled data on isolated phytoplankton and bacterial EPS to test whether biologically produced EPS could represent the bulk of the in situ Fe-L (Table 1). The maximal contribution to Fe-L was estimated for each ligand type (Fe-L_Carb, Fe-L_EPS; Fig. 4a) using their respective maximal iron-binding capacity (Table 1), as was done for the L_HA and L_FA. Further, the complexing capacity of each ligand type was calculated using the in situ excess ligands (eL = Fe-L − DFe) multiplied by their respective conditional binding affinities for Fe (Table 1). In theory, for a ligand to potentially influence the in situ Fe biogeochemistry, its

complexing capacity should be equal to or higher than that of the in situ Fe-L (eL × $K_{Fe'L}$; Fig. 4b). We note that such a calculation relies on similar assumptions as those for Fe-L$_{eHS}$. Hence, to be of significance for Fe biogeochemistry in the SO, an estimated ligand type should have a complexing capacity much greater than that of the in situ Fe-L. Carb should not play a major role in Fe biogeochemistry as their log $K_{Fe'L}$ are low[5,46] (Table 1). Moreover, their average complexing capacity is 3.8-fold lower than the in situ value, similar to what is observed for FA (Fig. 4b). Overall, Fe-L$_{EPSin}$ was similar to in situ Fe-L, while both Fe-L$_{EPSphyto}$ and Fe-L$_{EPSbact}$ were greater than Fe-L by two orders of magnitude (Fig. 4a). The most plausible explanation for these results is that EPS are not explicitly produced to bind Fe, in contrast to siderophores. Rather, EPS also bind numerous other trace metals[24,40–42]. Nonetheless, with only 1% saturation with Fe, EPS could represent the bulk of the in situ Fe-L. Given that in the EPS used here, Fe was present at a level largely exceeding zinc (by 2–15 fold) and copper (by 20–50 fold), this 1% Fe saturation is a robust approximation[40,41]. We conclude that the complexing capacity of EPS from phytoplankton and bacteria was high enough to regulate in situ Fe biogeochemistry and bioavailability (Fig. 4b). As such, our observations shed light on the nature of the marine humics controlling Fe biogeochemistry, showing that autochthonous EPS and biogenic humics can represent the bulk of the eHS and Fe-L across the SO, and most likely also in the South Pacific Ocean.

Because the eHS and Fe-L properties of in situ EPS are much closer to those of phytoplankton than bacterial EPS (Table 1), phytoplankton are likely the dominant source of the in situ EPS in surface waters. The relative contributions of phytoplankton and bacterial cells to total particulate organic carbon (POC) varied greatly across the SO. In the Mertz polynya, for instance, the absolute heterotrophic bacterial abundance was similar to that in other SO regions despite the much greater phytoplankton biomass (Supplementary Fig. 2), suggesting that bacterial EPS might be relatively less important in productive waters, with bacteria instead increasing their share of total EPS in oligotrophic waters.

## Discussion

It is widely acknowledged that both terrestrially-derived and marine humics are present in all marine systems[14,16–23,28,38,43]. Compiling our extensive SO dataset with previous data from other ocean basins, we show that applying the Fe complexing capacity of standard terrestrial humics to measured eHS fails to represent the bulk of iron-binding ligands in the SO and the South Pacific[14,19,23,24] (Fig. 3). Moreover, the lack of relationship between eHS and DFe:eHS for the southernmost regions (Fig. 2, Supplementary Fig. 3) implies a different nature/behaviour for in situ Fe-binding humics compared to other ocean basins. We note that this observation remains valid when DFe is compared to Fe-L (Supplementary Fig. 5), pointing towards a basin-scale difference in Fe-binding ligands that remains unexplained. Interestingly, a recent modelling study showed that the fate of Fe in surface waters is mainly controlled by biological Fe uptake and ligands for most of the Southern and Pacific Oceans[31], highlighting a larger pivotal role for biologically mediated Fe recycling in comparison to other ocean regions.

In the upper 250 m of the water column, the residence time of Fe ranges from sub-annual to annual[47], and (semi-)labile DOM typically turns over in hours to years[44]. Over the entire ocean depth, Fe residence time varies from <10 to 1000 years[48], while the average age of DOM is >5000 years and up to 20,000 years for the most refractory compounds[49,50]. These differences in residence time for DFe and DOM suggest that Fe-binding eHS are more labile than terrestrial humics, especially in surface waters. Like DOM, Fe is recycled through grazing[51,52], cellular lysis[53], and ligand production by phytoplankton and bacteria[9,11,12,41]. Laboratory[51] and field experiments[52] indicate that Fe recycling rates are rapid, typically occurring on time-scales of hours to

days, and therefore strongly control the Fe chemistry apportionment into the soluble and the colloidal fractions[51,54].

A compilation of studies of eHS concentrations and DFe:eHS ratios (Fig. 3) indeed reveals global depth-related differences in humics. In surface waters (0–100 m), eHS concentrations are greater ($75.7 \pm 78.7\ \mu g\ L^{-1}$, $n = 262$) than in deeper water (below 100 m; $39.9 \pm 34.2\ \mu g\ L^{-1}$, $n = 244$) but their level of Fe saturation (DFe:eHS) is greater at depth ($25.7 \pm 53.6\ \mu g\ L^{-1}$, $n = 236$) than at the surface ($11.8 \pm 10.8\ \mu g\ L^{-1}$, $n = 257$). The increase in Fe saturation levels of eHS with depth is consistent with previous observations[14,43] and suggests an essential role for eHS in the deep-ocean Fe inventory. At depth, DOM and possibly also Fe-L decrease along the global circulation pathway[55], suggesting that the remaining compounds relate to long-lived refractory Fe-L, including bacterially produced carboxyl-rich alicyclic molecules (CRAM)[26,32] and eHS. Unfortunately, waters deeper than 1000 m were not sampled in our study and, to date, no measurements exist for bacterial EPS at greater depths. CRAM are refractory DOM components that have been associated with a strong eHS signature and likely represent a significant portion of organic ligands at depth[43]. Indeed, CRAM and HS share many properties, including their ubiquitous distribution in the ocean[15,32], their origin as bacterial degradation products[26], their high content of carboxylic groups[15,32], and their role as Fe-L[14,23,43]. It is estimated that one-third of the refractory DOM pool consists of CRAM[49,50] and that bacterially derived CRAM can represent up to 50% of DOC at depth[25,32,49].

Our study suggests that EPS represent the bulk of Fe-binding marine humics and ligands, and are a key component for Fe biogeochemistry in the upper 1000 m of the SO water column. The biological production of eHS or HS has already been observed for bacteria[41], phytoplankton[41,43] and zooplankton[56] in surface waters, and by bacterial processing of DOM at depth[25,26,28]. Additionally, most of the other Fe sources to the SO, including dust[24,57], hydrothermalism[58], upwelling[14,43], coastal sediments[13,30], and ice melt[30,59] have associated (e)HS content. The Fe-binding capacity of these differently sourced eHS remains unknown and might differ substantially from the terrestrially-derived standards used. Indeed, the Fe-binding capacity for the EPS selected ($415–602\ nmol\ Fe\ mg^{-1}$ eHS, calculated from Table 1) was much greater than for the Suwanee River standards.

Recently, it was estimated that dust deposition may support >30% of primary productivity in the SO[60]. Considering the eHS content of atmospheric dust[24] ($3.7–5.8\ mg\ eHS\ g^{-1}$ dust) and dust deposition rates[60] ($0.2–12\ mg\ m^{-2}\ d^{-1}$), we estimate a daily eHS input to surface waters of $0.74–69.6\ \mu g\ eHS\ m^{-2}$. Considering that mixed layer depths (MLD) typically exceed 20–50 m in summer (and may extend to >100 m in winter)[61], the potential contribution of the aeolian eHS fraction to the total eHS inventory of the SO remains negligible. Although no constraints on Fe-L associated with dust deposition are available, it seems reasonable to postulate that regardless of the magnitude of the atmospheric Fe input flux, its interaction with in situ biogenic Fe-L (i.e., EPS and other Fe-specific ligands such as siderophores) determines the fate of Fe in surface waters, similar to hydrothermal Fe inputs[62].

EPS represent organic compounds excreted by organisms and suffer from a similarly loose definition to HS, likely due to their complex nature[41–43]. Here, we show that hydrolysable carbohydrates, a generally important constituent of EPS, cannot explain EPS's Fe-binding properties and contribution to Fe biogeochemistry in the SO. Indeed, EPS comprise more than carbohydrates – they are poorly characterised and polydisperse macromolecules for which conformation effects and the functional groups responsible for iron-binding remain unelucidated[40–42]. Yet, similarity to HS is expected, including potential roles for carboxylic, phenolic and catechol functional groups in Fe-binding[17]. The log $K_{Fe'L}$ of phytoplankton and bacterial EPS were similar to in situ values reported across contrasting regions of the SO ($11.28 \pm 0.46$, this study) and in the global ocean[63] ($12.34 \pm 1.04$,

considering the same competitive ligand as used here). Together, these results support the idea that EPS can represent the bulk of Fe-binding ligands.

Even though we used the best data available on biologically produced EPS and the marine eHS signature, our calculations do not reflect in situ Fe chemistry. Indeed, we assume that all measured eHS forms (HA, FA, EPS) behave like a single ligand type and that only Fe reacts with this ligand in a 1:1 Fe:L binding. These assumptions likely result in an overestimation of the concentration of any of the single ligand types considered. Additionally, as in most Fe biogeochemistry studies, we only report overall stability constants, yet HA and in situ Fe-L are likely polydisperse and possess different Fe-binding sites[37,41] with different Fe-binding conditional constants that become relevant at different DFe concentrations[37] and at different pH values[64]. Advances in our understanding of fine-scale Fe biogeochemistry are limited by data availability. Nonetheless, considering specific Fe-binding properties of different humic compounds, we show that with a realistic 1% Fe saturation, biologically produced EPS can account for the bulk of the in situ Fe-L while terrestrially-derived HA and FA cannot. Our data unequivocally identify phytoplankton- and bacterial EPS as biogenic marine humics forming most of the in situ Fe-L and therefore controlling Fe bioavailability in the surface SO, and possibly in other oceanic regions such as the South Pacific. Most marine process studies have been conducted with terrestrial or freshwater humic standards[14,18–23]. Although this standard approach allows for data intercomparability and provides a common baseline for eHS quantification[20,33], the lack of a marine humics standard is widely acknowledged as problematic[14,17,20,23]. Our data show that bacterial and phytoplankton EPS could represent a relevant marine humic material that is relatively easy to isolate and that can be used to elucidate marine Fe biogeochemistry and cycling. To advance further, large-scale efforts in isolating marine humics, either from relevant cultures or at sea, would be needed, as would a detailed characterisation of a relevant marine humic standard, including its polyfunctional nature relevant to Fe-binding[37] and ligand exchange kinetics[38].

In this study, we identified microbial EPS as the key marine humics controlling Fe biogeochemistry in the surface waters of the SO. Additionally, given the limited measured variability in eHS, Fe-L and log $K_{Fe'L}$ for a given EPS type, the respective contributions of phytoplankton and bacterial EPS could be discussed. In surface waters, because phytoplankton EPS dominate the in situ EPS pool, one can expect that they modulate Fe bioavailability and relieve Fe and organic carbon (co)-limitation of primary producers[6,7] and heterotrophic bacteria[65,66]. However, bacteria also produce EPS that can bind Fe in a form that is highly bioavailable to phytoplankton[40]. Furthermore, through their remineralisation of organic molecules[25–28], bacteria release weak Fe-L[65] that can benefit phytoplankton[40,43,65] and thus close the Fe cycling loop in surface waters. Our data suggest that bacteria abundance (and by extension, activity) sets eHS levels at depths greater than 100 m, where a correlation between them was observed (Supplementary Table 3). At depth, HS and eHS composition is regulated by bacterial remineralisation of convected surface DOM and sinking POC[28], with implications for the deep Fe inventory and global Fe cycling. Interestingly, EPS have the ability to change conformation with time[42] and form gel-like matter such as transparent exopolymer particles and marine snow, which are recognised as important particles for carbon export[67]. These large organic-rich particles are known to absorb Fe, including small inorganic Fe colloids, and potentially contribute to Fe scavenging at depth[31,48]. The role of EPS in connecting the Fe and carbon cycles from the surface to the deep ocean might thus be even greater than reported here. However, a lack of data on deep-water bacterial EPS and a poor understanding of the contribution of EPS to transparent exopolymer particles at the surface and depth prevents further assessment.

Our identification of phytoplankton and bacterial EPS as key marine humics and Fe-binding-ligands across the SO reveals fundamental interactions between the biological carbon pump and Fe cycling. In the SO, the biological carbon pump is currently underestimated[68], and projections of future primary productivity are dependent on how we model its dependence on Fe and light availabilities[10]. Our findings argue for a revision of how we understand and model primary productivity and carbon sequestration in the SO, a region responsible for an important share of the global ocean response to current and past climate changes.

## Methods
### Water sampling
Sampling was conducted onboard the R/V Akademik Tryoshnikov from 20th December 2016 to 22nd February 2017 (Fig. 1, Supplementary Table 1). Water for trace metal and ligand measurements was sampled using an autonomous rosette (Model 1018, General Oceanics, USA) on a Dyneema line equipped with acid-washed Teflon-coated 10 L Niskin X bottles, with bottles manipulated in a clean container under an HEPA filter as described in ref. 69. Water from 0 to 1000 m was filtered by gravity through 0.2 μm acid-washed capsule filters (Acropak 200, Pall). Sampling and manipulation related to Fe biogeochemistry followed GEOTRACES guidelines[70]. Water for the biological parameters was sampled using a 12 L Niskin bottle deployed on a rosette equipped with a Seabird 911 CTD[71]. Both rosettes were deployed within a 1 h time window to collect comparable data. Macronutrient data from the two deployments showed a similar water column structure[72].

### Dissolved Fe concentrations
Samples for trace-metal concentrations were collected in LDPE bottles acidified with $HNO_3$ to pH <1.8 and stored for at least 6 months before analysis. Laboratory work was carried out under trace-metal clean conditions in ISO 5 clean hoods, using ultrapure reagents. Two different analytical methods and sample sets were used, and they showed good accuracy and analytical metrics[35,69,73]. Briefly, dissolved Fe concentrations for Leg 1 (Cape Town to Hobart, Fig. 1) were determined at ANU by isotope dilution. Samples were pre-concentrated, and the seawater matrix was removed (Nobias PA1 resin) offline using a home-built, automated pre-concentration system, and then eluted in 1 M $HNO_3$. Weighed sample aliquots were spiked with enriched isotopes ($^{57}Fe$) and internal standards (Sc, In, Yb), then analysed by inductively coupled mass spectrometry (ICP-MS) on either a Neptune or Element XR instrument (both ThermoScientific).

Dissolved Fe concentrations for Leg 2 (Hobart to Punta Arenas) were measured using samples taken for metal stable isotope composition, which were acidified and amended with Fe double spikes. Metals were extracted using Nobias PA1 resin and purified using AG-MP1 resin, following previously published methods[35,69]. Fe concentrations and isotope ratios were analysed at the University of South Florida using a Neptune with an Apex-Omega desolvating system, a Jet Ni sampler cone and an X-type Al skimmer cone.

### Electrochemistry
Electrochemical measurements were made as per[9] using a 0.52 mm² hanging mercury drop electrode and stirring during deposition steps using a rotating PTFE rod set to a stirring speed of 3000 rpm. Analyses were conducted in a laminar flow cabinet (600 PCR workstation, Air-Clean Systems) at ambient temperature.

**Fe(III) chemical speciation.** The Fe speciation was determined by a CLE-AdCSV technique as in[9,74]. Briefly, a seawater aliquot was buffered to pH 8.2 with a 1 M borate buffer to which 0–9 nM Fe was added. After 1 h equilibration at room temperature, 5 μM salicylaldoxime solution

(SA, Acros Organics, competitive ligand) was added to all the tubes. Following overnight equilibration, the analysis was conducted using the following parameters: 30 s air purging, 200 s deposition time at 0 V. Voltammograms were analyzed with ECDSoft[75], considering the 4th derivative with a tangential baseline, and the software ProMCC[76] was used to analyse the resulting Fe-binding ligand titration data and quantify both the total concentration of in situ ligands (Fe-L in situ) and the conditional stability constant ($K_{Fe'L}$) of the complexes. Parameters were calculated assuming the detection of one ligand class and considering van den Berg fitted values.

**Electroactive humic substances (eHS).** Electroactive HS was determined using the voltammetric method from[33] with standard addition

SRFA (standard 1, IHSS, referred as FA), hydrolysable carbohydrates (Carb) and biologically produced EPS (phytoplankton, bacteria and in situ) were selected. First, the in situ measured eHS was transformed into the selected humics concentration using their specific eHS content (Table 1). Considering the average maximal Fe binding properties (Table 1) for each humics type (HA, FA, Carb, and EPS), the corresponding Fe-binding ligand concentration was calculated assuming that all eHS behave as a specific ligand type ($L_{HA}$, $L_{FA}$, $L_{Carb}$, and $L_{EPS}$). Their contribution to Fe-binding ligands was then obtained by comparing $L_{HA}$, $L_{FA}$, $L_{Carb}$, and $L_{EPS}$ with in situ Fe-L. Overall, this approach followed calculations made previously[19,21,22]. For example, to estimate the contribution of EPS to in situ Fe-L, one applies the following formulae:

$$\text{Equivalent concentration of } EPS\left(mgL^{-1}\right) = \frac{eHS\left(mgL^{-1}\right)}{eHS \text{ content of } EPS(mg\ eHS\ mg^{-1}EPS)} \tag{1}$$

$$L_{EPS}\left(nM\right) = \text{equivalent concentration of } EPS\left(mgL^{-1}\right) \times Fe-L \text{ content of } EPS\left(nmol\ Fe\ mg^{-1}EPS\right) \tag{2}$$

$$\text{Potential contribution to in situ ligands}(\%) = \frac{L_{EPS}}{Fe-L} \times 100 \tag{3}$$

of SRFA (IHSS, std 1, 0–90 μg L$^{-1}$)[9]. This method is based on CSV and makes use of the adsorptive properties of Fe–HS complexes on the mercury drop electrode at natural pH. As such eHS can be described as the fraction of HS binding Fe that is electroactive. A mixed reagent solution (KBrO$_3$, EPPS, NH$_4$OH, 750 μl) was added to 100 mL of seawater in order to obtain a pH of 8.2, along with 20 nM of inorganic Fe (ICP standard, Sigma) to saturate the in situ eHS. After 2 h in the dark at room temperature, analysis was done using a 300 s nitrogen purge time (PanGas) and 200 s deposition time. Detection limit was 1.9 μg L$^{-1}$ SRFA equivalent and accuracy was within 3% for a solution of 95.0 μg L$^{-1}$ SRFA[77].

#### Carbohydrates
Hydrolysable mono- and polysaccharides are considered as labile organic compounds that were quantified by 2,4,6-tripyridyl-s-triazine spectroscopy[78]. D-Glucose was used as a standard. The instrumental detection and quantification limits were 2.46 μM C and 4.29 μM C, respectively[79].

#### EPS and in situ, DOM selected as marine humics
Most of the EPS selected are relevant to the SO and its low iron levels; all EPS are of marine origin or from marine strains. The bacterial strains included *Pseudoalteromonas sp.* CAM025 isolated from Antarctic sea ice[41], *Cobetia marina* L$_6$[9,80], and *Vibrio alginolyticus (Hassler, Beaudoux, unpublished data)*. Two separate isolations of *Cobetia marina* L$_6$ were analysed. The phytoplankton strains included Prymnesiophytes from the Australian National Culture Collection, *Phaeocystic antarctica* (CS 243, Prydz Bay, Antarctica) and *Emiliania huxleyi* (CS 812, Mercury Passage, Tasmania, Australia). The in situ EPS or DOM was isolated from the North Atlantic[43] (1000 m depth, 35°39.8′ N 74°30.8′W) and from the Sub-Antarctic Zone[40] (depth of fluorescence maximum, 25 m, 159°5 E 46°2 S) coincident with a coccolithophorid bloom[40] (0.69 μg L$^{-1}$ Chl *a*). The respective contributions to these selected organic materials of hydrolysable carbohydrates, eHS and Fe-L were measured in inorganic trace-metal clean seawater[9,41].

#### Data analysis
To estimate the potential contribution of different humics to in situ Fe-binding ligands in the ocean, SRHA (standard 1, IHSS, referred as HA),

To further assess the potential for each selected humic to impact SO Fe biogeochemistry, we calculated the complexation capacity using the product of excess ligands ($eL = Fe - L_{in\ situ} - DFe$)) and multiplied it by the conditional stability constant of each ligand type. For a ligand to potentially influence SO Fe biogeochemistry, its average complexing capacity should equal or exceed the in situ complexing capacity given by $eL \times K_{Fe'L}$. Data are discussed with respect to different humics types and support investigation into the respective roles of bacterial and phytoplankton EPS.

Statistical relationships amongst different parameters were investigated using ACP and a Pearson correlation table using Sigma-Plot ver. 14 with a statistical significance set at the level of 0.05.

### Data availability
The ACE data from this study has been deposited in the Zenodo database as part of the Swiss Polar Institute data management plan under accession codes 3247384, 3247383, 2616606, 3634411, 3897170, 3250136, 2635686. The data generated for global ocean comparison in this study are provided in the Supplementary Source Data file. Source data are provided with this paper.

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

## Acknowledgements

We warmly thank the captain, crew and water sampling team of the RV Akademik Tryoshnikov as well as the chief scientist, the late David Walton. We are also grateful to D. Cabanes, M. Sieber, T. Conway, S. Trimborn, T. Brenneis, M. Fourquez, and L. Stirnimann for data analysis, D. Antoine, S. Thomalla, and C. Robinson for data on pigments and Fv/Fm, M. Zamanillo for flow cytometry samples, H. Forrer, R. Flynn, M. Fourquez, H. Little, M. Zamanillo, P. Cortés-Greus, and P. Rodríguez-Ros for help with sample collection, and C. Boisset, A.-C- Baudoux, F. Lelchat, and C. Mancuso Nichols for EPS isolation. ACE was a scientific expedition carried out under the auspices of the Swiss Polar Institute (SPI), supported by funding from the ACE Foundation and Ferring Pharmaceuticals. C.H. and S.L.J. were supported by the Swiss National Science Foundation (PP0OP2_166197 and PP0OP2_172915, respectively); C.H. is supported by the Ferring Pharmaceutical – Margaretha Kamprad Chair in environmental sciences attributed to Prof. J. Chappellaz, S.E.F. received support from the South African National Research Foundation (129232 and SANAP230527110611). S.L.J. holds a Consolidator Grant from the European Research Council (ERC-2018-CoG #819139). R.S. holds an Advanced Grant from the European Research Council (ERC-2018-AdG #834162). The ICM-CSIC is supported by a "Severo Ochoa" Centre of Excellence grant (CEX2019-000928-S) from the Spanish government.

## Author contributions

C. Hassler designed the study and interpreted data, C. Hassler, S. Jaccard and M. Ellwood secured funding; C. Hassler, S. Jaccard, R. Simó, S.

Fawcett and M. Ellwood collected samples, participated in data analysis and in the manuscript preparation.

## Competing interests

The authors declare no competing interests.
