## [Peer Review file · Nature Communications]

Marine biogenic humic substances control iron biogeochemistry across the Southern Ocean

Corresponding Author: Dr Christel Hassler

Version 0:

Reviewer comments:

Reviewer #1

(Remarks to the Author)

This paper presents important new data on the ligands for iron from the Southern Ocean, an under-sampled region, and provides insight into identifying those ligands.

Strengths:

The strength in this paper lies in new data for ligands, humic substances and EPS from the Southern Ocean, an iron-limited region that is under-sampled and poorly understood. This is an important topic that deserves attention. The additional analyses into identifying the sources of the ligands are an important contribution. This study builds on previous work by the lead author into the importance of EPS as a complexing agent for iron in the Southern Ocean, and points to a biogenic source of ligands. The general assumption within the scientific community would be that most of the ligand pool in the upper Southern Ocean is biogenic, and therefore these findings are not necessarily a surprise and thus the narrative is a little off the mark, although that shouldn't discredit the importance of this new work in providing better insight into ligand sources and identity.

Concerns:

This paper struggles with some key concepts and is weak on getting across its true importance, which is crucial for an article in this journal. It lacks discussion of many other key works on this topic, which for some reason are omitted from the 'global ocean' data in figure 3, yet which includes data from less relevant studies. For example, there is no discussion of a study that already reported biologically mediated production was the main source of eHS in the Mediterranean, a nearly landlocked basin and therefore arguably more surprising to not be dominated by terrestrial humics (Dulaquais et al., 2018), or of work in the Arctic as a contrasting polar region (Slagter et al., 2019; Slagter et al., 2017). Instead, the justification for this work appears to be that the findings here are unexpected when considering a single previous study by Whitby et al. (2020), yet the conclusions of that study appear to have been misrepresented in order to fit the narrative here.

This paper needs to be written more accurately and transparently to best place these findings in the context of existing research. The paper should be revised to better emphasize that its novelty is in answering questions posed by previous studies to enhance our understanding of this little understood topic, and in studying this important region.

The paper would also benefit from more clarity for the non-expert on what the different ligand types are, and how they are related, particularly an explanation of the crossover between EPS and humics, as well as more detail in the methods section and a more honest view of any assumptions and limitations.

Background info

Firstly, relevant for lines 50-55, 196 and elsewhere:

'Humic substances' is a catch-all term which represents a continuum of organic compounds following degradation of (terrestrial or aquatic) biomass, including the degradation of polysaccharides, proteins, lipids, nucleic acids, lignin, cutin, melanin etc. Within humic substances, humic and fulvic acids are merely operationally-defined fractions, with humic acids soluble in natural and alkaline solutions only, and fulvic acids soluble at all pH levels. EPS is not a separate operationally defined fraction of humic substances (as inferred in line 196).

By definition, all soluble humic substances are a mixture of both HA-like and FA-like material; i.e. all samples contain a mixture of some ratio of material that either is soluble at all pH, or is not. HA and FA cannot empirically be distinguished from one another using the voltametric method referenced. eHS is itself an operationally defined fraction (electroactive humic-like material), and thus also a mixture of compounds that are either soluble at all pH (FA-like) or are not (HA-like). Any EPS that is detected by the eHS method would fall into either the HA-like or FA-like category purely because of this operational definition, as will many other compounds, not be distinct from it - it cannot be soluble and not be one of these if it contributes to the humic pool. The fact that some natural EPS in a sample behaves electrochemically similarly to the added FA used for the standard addition and is thus caught up in the measurement demonstrates that it shares FA-(or HA)-like properties. One can either argue that humic substances do not include EPS and thus this is an interference, or (some) EPS are HA/FA, but both cannot be true, as the authors appear to suggest.

Finally, the eHS method cannot distinguish between sources of eHS, and cannot alone inform of the presence terrestrial or marine material, regardless of the standard used for quantification.

Literature:

There are many other relevant works that the authors may wish to consider for a broader understanding of this topic and in building their narrative, and the recommended expanded global eHS dataset for Figure 3; it is not necessary for all of these to be included in this article (not all have the relevant data for the figure), but I share in case they are of relevance (Dulaquais et al., 2023; Dulaquais et al., 2018; Fourrier et al., 2022; Laglera et al., 2019; Mahieu et al., 2024; Muller, 2018; Slagter et al., 2019; Slagter et al., 2017; Sukekava et al., 2018; Sukekava et al., 2024; Whitby et al., 2020a; Yamashita and Nishioka, 2023; Yamashita et al., 2020).

Comments

Title is appropriate

L17-18 This would benefit with a sentence introducing what humic substances are before discussing their relationship with iron.

L18 – this statement as written is incorrect, or the citation used does not support the statement.

*(Note that reference 6 in the bibliography also has a typo: Whitby, H. et al. humic-like 'substances')

The Whitby et al paper does not propose that terrestrial humics dominate the ligand pool globally. There is a suggestion that the North Atlantic specifically has a higher degree of terrestrial influence, but not that terrestrial humics form the majority of iron-binding ligands in general. There is nothing to suggest terrestrial humics would be expected to dominate the ligand pool in the surface of the Southern Ocean, and I don't believe the community think this is the case.

See figure 4 in Whitby et al which specifically separates iron limited from terrestrially-influenced regions, and the following relevant excerpts from the Whitby et al. study:

'Higher concentrations at low salinities demonstrate a terrestrial source, as would be expected. However, elevated concentrations in mesopelagic waters and relatively consistent values at open ocean locations also agree with a marine source and a refractory component [of humic substances]'

'When considering that a large fraction of organic matter in the deep ocean is aged and allochthonous¹⁰, our results suggest that the persistence of terrestrial humics in the deep ocean may be pivotal in the global ocean iron cycle. However, there are many sources of both iron and humic material to seawater and it is likely that aged humics in the deep ocean are derived from a combination of terrestrial³⁴, marine³⁷ and even hydrothermal sources¹⁶. Regardless of origin...'

'Thus unsurprisingly, the humic contribution to iron complexation is dependent on the concentration of other, stronger ligand groups, varying regionally and with depth.'

'Deep waters reflect an accumulation of aged humic material of higher aromaticity^{28,29}, linked to the persistence of terrestrial humics^{32,34}, the selective decomposition of organic matter³⁵, and the hypothesized aging process of humification^{24,28,34}.'

Considering all of that, I think it necessary to reconsider the justification for this study and the text alluding to this within the abstract and other parts of the paper.

E.g. the authors may consider something more transparent, along the lines of:

Based on iron stoichiometry with electroactive humic substances, it has been proposed that humic material composes an important fraction of the iron-binding ligands in the ocean^[6]. Humic substances are [definition]....., originating from a variety of sources. Some regions have a strong terrestrial humic influence, such as the Arctic Ocean^[refs]. However, there is very limited data on humic ligands in the Southern Ocean, which is characterized by.....

L20-22 Humics do not account for most of the ligands, but EPS account for most of the humics and most of the ligands? Is this because only a small fraction of EPS is picked up by the humic measurements? Please clarify the wording here.

L43 – how does this work fit in with the recent study (in Nature) that claims that the ligand pool is less important than previously thought and that most iron in seawater may instead exist in authigenic colloids? (Tagliabue et al., 2023).

L50-55 states:

'It has recently been proposed that Fe biogeochemistry across the global ocean is regulated by humic substances (HS, resulting from the degradation/humification of bacterial, plant and animal matter in soil, sediments and natural waters)⁶. However, the study considered only terrestrially derived humics (Suwanee River fulvics and humics, further referred as HA and FA) as the main contributors to electrochemically detected iron-bound HS (eHS), despite other sources like atmospheric dust deposition¹⁵ and microbial reworking of marine DOM¹⁶⁻¹⁸ having been reported.'

The eHS method relies on a standard for quantification of eHS concentration in seawater; typically, Suwanee River Fulvic Acid or Humic Acid isolates are used, which are commercially available from the International Humic Substances Society. While a marine standard (and improved extraction procedure) would be preferable, currently there is not one commercially available at scale and the IHSS standard allows different studies to be compared with one another (as the authors do here). Terrestrial and marine humics share many similar components, and a terrestrial standard does allow quantification of the natural eHS, which in seawater is likely composed of some ratio of terrestrial and marine-derived humic material ('humic' encompassing many different things). It does not infer through its use that all of the material is of terrestrial origin. Furthermore, all humics and thus we can only assume all eHS is a variable mixture of FA-like and HA-like material, which cannot be determined using the existing method.

The cited study uses these known operational definitions of HA and FA to explain apparent trends regionally and with depth, and to infer the likely source of the material, such as using aromaticity from fluorescence measurements to infer changes to binding capacity, based on what is known for the FA and HA standards. However, I don't believe the cited paper suggests that the FA and HA like components exclude EPS (or any of the many other components of humic substances). As stated above, EPS is not an accepted operationally defined fraction separate to FA or HA. Neither the paper cited (Whitby et al. 2020) nor this paper can empirically distinguish how much of the eHS is terrestrial, or how much actually is FA-like or HA-like. Just because an FA standard is used does not mean all of the eHS is FA, and vice versa, as the natural eHS peak can be quantified with either standard.

Furthermore, as noted above the cited study discusses a variety of sources of humics, not only terrestrial, and thus the discussion is again incorrect.

L59 – add space contrasting regions

L68-69 please clarify what is meant by 'how the correlation between DFe and eHS that served as a foundation for this assertion is not seen (Figure 2)'. Figure 2A appears to show data from citation 6, where the data from the Southern Ocean also appear to be in a similar position to the data here in panel B – i.e. low DFe, high eHS concentrations, unless I am missing something? (noting that Figure 2A is challenging to read when using triangles for every basin, even if they are different orientations. I recommend improving the figure by changing the icons). Is the point being made here that there are some values where DFe exceeds the humic binding capacity, suggesting that other ligands are important? If so, this is interesting and needs to be stated more clearly, although if that is the intention, it does not fit with the next two lines of text that suggest that DFe concentrations were low but eHS was similar to concentrations found elsewhere (lines 71-72). Clarity needed for this section.

L77 – why Liverpool Bay, which is coastal, and the Western Antarctic study, which does not have ligand data, yet many other more relevant studies are omitted? There needs to be justification for the data used in this figure – and similarly, why other studies are ignored. I strongly urge the authors to include a wider range of published humic studies in order to have a transparent and comprehensive comparison that actually is an update on the original paper. See comment under Figure 3 below.

L99 – It should be noted that the percentage contribution of eHS to FeL here is similar to findings in the South Pacific (Cabanés et al., 2020; Dulaquais et al., 2023).

L118 – reworking of DOM, and also of POM - see 'Contribution of electroactive humic substances to the iron-binding ligands released during microbial remineralization of sinking particles' (Whitby et al., 2020a). It may be relevant elsewhere in this paper too as it includes data from the Southern Ocean.

L126 – how is the amount of ligand that is allochthonous determined? (and similarly elsewhere, the amount of eHS that is terrestrial?) or is this purely through order of elimination when using measurements of known biogenic components? Clarity needed here.

L129 – add space before 'Whereas' and then either reword this sentence so it is a complete standalone sentence, or join it with the subsequent sentence '...phytoplankton, carbohydrates...' – there is also a typo there and the word should be 'constitutive'.

L193 – add 'is what determines'

L205 – 207 reword this section to better convey the true novelty and importance. Remove revisit – see above, there is nothing to revisit, just build on. What is important here is that we have new data for the Southern Ocean, and further evidence that eHS has a strong biogenic component, supporting previous studies e.g. Dulaquais et al (2018). The authors could use this work to better clarify the relationship between EPS and HS.

One could argue, if there is a lot of excess ligand here, and a similar amount of humic material here to other regions, it is not

then purely the low iron supply that is limiting productivity, and the ligands themselves have little impact on iron geochemistry, no matter what they consist of? The message of why this study is important does not come across as powerfully as it should for a Nature Communications article.

I recommend the authors try to emphasise why it is important to know that most of the eHS present is of biogenic origin and thus explain the true novelty of this study.

Figures

Figure 1 is clear and well explained. However, I think it is a shame not to have the map shown as extended fig 1 within the main text.

Figure 2A – use different symbols instead of all circles – even in different orientations, they are still challenging to distinguish between (and maybe more so for those with visual disabilities or dyslexia for instance).

Figure 3 - In general I like this figure, but as currently presented it is misleading as it cherry-picks only some published humic data to support a certain narrative, omitting other data, and with incorrect labels.

In panel C, 'Global Ocean' should either read 'North Atlantic Ocean', as according to the paper, only the samples where both humic and ligand data was available was used for this percentage which was only the N Atlantic, or actually be a compilation of the existing global data that is available. I strongly urge the authors to do the latter.

For example, multiple studies have shown eHS to compose around 20%–60% of the iron-binding ligand pool (Dulaquais et al., 2018; Fourquez et al., 2023; Whitby et al., 2020b), while iron-humic complexes can be the dominant form of dissolved iron in the terrestrially-influenced Arctic Ocean (Laglera et al., 2019; Slagter et al., 2019). There is also another study in the Pacific which corroborates the Cabanes study with low eHS relative to DFe (Dulaquais et al., 2023). Combining these would give a more comprehensive global dataset. I therefore encourage the authors to either separate the data by basin, or, to have a Southern Ocean and 'Global' or 'Rest of ocean' dataset, to be truly transparent.

Figure 4, Table 1, and section L135 -169 – This section and the linked visual material is challenging to follow, particularly considering line 150 'Together, this shows that EPS ability to bind Fe cannot be predicted from their hydrolysable carbohydrate content'. This section would be improved by a simple explanation for the lay audience (not EPS/ligand or DOM experts) explaining what is being compared and how they are related, particularly relating to the different forms of EPS and how they relate to eHS. Out of curiosity, how do EPS compare to other humic measurements, such as fluorescence measurements?

Methods

I hoped the methods section would give a more detailed explanation of some concepts that are difficult to follow in the main text, but it is rather sparse and missing some key details.

General:

Ensure throughout that units are correctly written, e.g. μg instead of ug.

There is confusion with references – for example, on line 477 in methods, quotes reference 60, but there are only 19 references in the methods and 50 in the main text, so which is reference 60?

This may be up to the journal, but with the separate bibliography and numbering system restarting in the methods, there are then some duplicate references where a single paper has two numbers depending on the section being read, which can make it confusing for the reader and possible to accidentally read the wrong bibliography and use the wrong reference. Is it possible to have a continuation of the original numbered reference list in the methods so each paper has only one number?

It looks like there is the inclusion of some methods for data that is not shown or discussed in the main text; depending on the rules of the journal, I would recommend moving any methods relating to only supplementary figures and tables to the supplementary info so that space can be saved in the methods to better explain some of the key parameters.

Specific:

L393 - missing some punctuation?

L401 – HNO₃

L420 – remove stop after 'as per 6'

L422 – remove comma after 'done'

L428 – only 30s purge?

L430 – add space 'and the'

L432 – why was only one ligand class fitted when SA typically gives multiple classes? It would have been interesting to see if two ligand classes were given and to compare the log K values of L1 and L2.

L434 – why was the Laglera and van den Berg 2009 method used and not the updated method (Sukekava et al., 2018)?

What is the full code for the IHSS standard; was the FA standard saturated with Fe prior to use? How many standard additions were performed? How was the detection limit determined?

L474 – change 'than in' to 'as in'. Typo in 'largely'

L476-479 Wording here is unclear. Used in situ ligand data instead of DFe but for what? They are different things.

Comparing to L gives an estimate of the % composition of L as eHS (ignoring that L is controlled by the artificial ligand selected, the detection window, pH buffered to etc). Comparing to DFe gives the relationship between the eHS and DFe, in turn ignoring that eHS can bind other metals and that DFe can be bound to other ligands. Both require assumptions, both are

useful comparisons that give you an insight, but insights into different things and as written this section does not make sense. It's impossible to check the reference 60 as it doesn't exist in the bibliography.

L477 – space between a potential

L480-482 – again this whole section needs to be written more clearly. 'we were able to transform measured eHS in an estimated Fe binding ligands concentration' does not make sense grammatically. Also, please detail how this was done, give references/calculations/assumptions etc.

L489 – 'should be equal to, or exceed, the'

L490 – 'with respect to different ligands' and 'investigation into the respective roles'

See attached for bibliography.

Reviewer #2

(Remarks to the Author)

The manuscript entitled "Marine Biogenic Humic Substances Control Iron Biogeochemistry across the Southern Ocean" presents a comprehensive investigation into humic substances, with particular emphasis on extracellular polymeric substances (EPS) as a crucial component. Although only 1% of iron appears to be complexed with these substances, the study is original and makes a significant contribution to the field by addressing a critical gap in the current literature. The importance of exopolysaccharides in the pool of iron ligands, as well as their substantial influence on the humic fraction, is well-documented. However, this paper effectively demonstrates, through a large dataset, the specific influence of these substances on iron biogeochemistry.

While the authors have successfully identified and addressed this research gap, some methodological and analytical choices require further consideration. Specifically, the use of correlations with low p-values but weak r-values suggests only a modest influence between the variables, rather than a strong correlation. Although the relationship is not random, the weak correlation raises concerns about the strength of the inferred relationships between the parameters.

Although the data presented are robust and the experimental design is generally well-structured, the reliance on these correlations may not sufficiently support the accuracy of the data analysis. It is recommended that the authors minimize the emphasis on these correlations as currently presented.

While the methodologies employed are appropriate for addressing the research question, it is unfortunate that the authors did not utilize a more recent version of the method for humic substances, such as the updated approach described by Laglera et al. (2007). The use of a more current methodology could have provided additional insights, particularly given the reliance on eHS results. Nevertheless, this does not detract from the overall quality of the study, which remains well-structured and informative.

In conclusion, I recommend acceptance with minor revisions.

GENERAL OVERVIEW

The paper provides a good explanation of how ligands and humic substances interact. However, the explanation is not always clearly presented, leading to some confusion in certain sections. I strongly suggest reading papers that describe the kinetics of humic substances with other ligands, as these could help address some of the information gaps.

Recommendations and Questions

-Line 28: Change "de facto" to "in practice."

- Line 54: The paper should not focus exclusively on SRFA and SRHA as the only methods for measuring humic substances. This sentence appears too restrictive. The current phrasing implies that these are the only approaches, which is not accurate. Consider rewording to allow for alternative methods in the study of humic substances.

- Line 59: Change "contrastingregions" to "contrasting regions."

- Line 61: Despite the significant p-value, the correlation is so small that it is difficult to believe any inverse relationship exists.

- Line 65: Did you extract the percentage of eHS to represent binding capacity? Since you transformed humic substances into eHS using the binding capacity of SRFA and SRHA, these percentages could help reinforce the strong positive correlation.

- Line 68: There are many other papers that describe this, such as Laglera et al. (2019). Be sure to reference relevant literature in the discussion, including Laglera et al. (2019), which provides additional insights into this topic.

- Line 69: It is unfortunate that the method was not updated, as using a more recent methodology would have increased the value of this data. By applying the updated approach, it would have been possible to observe the amount of iron complexed with humic substances and to calculate the percentage of iron bound to humic substances. Despite this limitation, I agree

that deep-water data could have had a greater influence on this outcome. Please reference the relevant paper that describes this issue.

- Line 70: The comparison method in Figure 2b seems unnecessarily complex, despite the values being similar. I suggest changing the triangle symbols to different shapes to improve clarity. Adjusting the graphical representation may help the reader interpret the data more easily, without introducing unnecessary complexity.
- Line 77: I disagree with the comparison of Liverpool Bay to other oceans. It would be more appropriate to use data from regions with similar environmental conditions or from areas with well-known concentrations of humic substances, such as the Arctic Ocean. The Arctic demonstrates high concentrations of humic substances and has a significant influence on the ligand pool. Liverpool Bay, with its distinct riverine inputs from major rivers such as the Alt, Clwyd, Dee, Ribble, and Mersey, and unique circulation patterns, is less comparable to open ocean environments.
- Line 87: Change “theFe-binding” to “the Fe-binding.”
- Line 95: This could be associated with the presence of other ligands. It is well known that if DFO binds to iron before humic substances, the latter cannot displace it. This explanation should reference Sukekava et al. (2024) and the concept of 'first come, first served.' A discussion on ligand competition, particularly about existing literature, would enhance this explanation.
- Line 102: The examples provided do not mention the Southern Ocean. I suggest reformulating the sentence to clarify that humic substances have been identified in various regions globally. A clearer description of the global distribution of humic substances would improve the reader's understanding.
- Line 133: There is a need to reinforce the importance of conducting kinetic studies to address uncertainties about EPS and their classification as weak ligands. This could be an important measurement for future studies.
- Line 189: The data is particularly interesting, as no riverine input seems capable of explaining the results. Dust and autochthonous humic substances appear to be the only plausible sources.
- Line 390: I assume all the materials for sampling and analysis were prepared following the GEOTRACES protocol. Should this be explicitly mentioned in the paper?
- Line 412: Has the zinc concentration data been utilized at any point in the analysis? If not, is it necessary to describe it in the text?
- Line 433: Why were calculations performed with only one ligand class? Could this be clarified further?
- Line 477: The reference appears to be incorrect; I could not locate any reference numbered 60. Please verify this.
- Line 482: I am unclear on how you calculated the different ligand concentrations for humic substances and carbohydrates. Could you provide a more detailed explanation of this section?
- Extended data Figure 1: While visually appealing, it is not clear how this figure contributes to the overall paper. It may be more effective to display only the sample locations without including the nitrate variability.
- Extended data Table 1: If all the sampling locations are already included in Figure 1, I would recommend omitting this table, as it may be redundant.

Extended data Figures 2 and 3, I recommend placing greater emphasis on the distinct behaviors observed across various locations, rather than focusing primarily on correlations between variables. Different regions exhibited unique patterns, particularly in ligand distribution. For instance, the High Nutrient Low Chlorophyll (HNLC) region and the Mertz Polynya displayed distinct groups of carbohydrates and eHS. These differences could be better integrated to highlight the varying chemical influences across different water masses, especially considering the influence of differing oceanic currents, as described in Smith et al. (2022) and Ardiningsih et al. (2020, 2021).

Sukekava, C. F., Downes, J., Filella, M., Vilanova, B., & Laglera, L. M. (2024). Ligand exchange provides new insight into the role of humic substances in the marine iron cycle. *Geochimica et Cosmochimica Acta*, 366, 17-30. <https://doi.org/10.1016/j.gca.2024.05.002>

Laglera, L. M., Sukekava, C., Slagter, H. A., Downes, J., Aparicio-Gonzalez, A., & Gerringa, L. J. (2019). First quantification of the controlling role of humic substances in the transport of iron across the surface of the Arctic Ocean. *Environmental Science & Technology*, 53(22), 13136-13145. <https://doi.org/10.1021/acs.est.9b04208>

Ardiningsih, I., Sander, S.G., Gerringa, L.J.A., Laan, P., & Middag, R. (2020). Sources of Fe-binding organic ligands in surface waters of the western Antarctic Peninsula. *Marine Chemistry*, 226, 103855. <https://doi.org/10.1016/j.marchem.2020.103855>

Ardiningsih, I., Seyitmuhammedov, K., Sander, S.G., Stirling, C.H., Reichart, G.-J., Arrigo, K.R., Gerringa, L.J.A., & Middag, R. (2022). Fe-binding organic ligands in coastal and frontal regions of the western Antarctic Peninsula. *Frontiers in Marine Science*, 9, 883482. <https://doi.org/10.3389/fmars.2022.883482>

Smith, P., Thompson, L., Williams, J., & Baker, A.R. (2022). Identifying potential sources of iron-binding ligands in coastal Antarctic environments and the wider Southern Ocean. *Frontiers in Marine Science*, 9, 948772. <https://doi.org/10.3389/fmars.2022.948772>

Reviewer #3

(Remarks to the Author)

Review of "Marine biogenic humic substances control iron biogeochemistry across the Southern Ocean" By Christel Hassler et al., submitted to *Nature Communications*, 2024

Scope of the manuscript, general assessment and recommendation

The manuscript by Hassler et al. is a contribution to the long-standing and biogeochemically very important question concerning the nature of the organic ligands that have been shown to control iron residence time and bioavailability in the ocean. It does so based on a new data set of measurements of electroactive humic substances, iron binding ligands, and dissolved iron concentrations mostly from the upper 200m depth below the surface in the Southern Ocean. It is argued that the bulk of the electroactive (iron-binding) humics and ligand iron binding in their samples is based on exopolymeric substances produced by bacteria and phytoplankton, as part of the marine humic pool.

Both the data set and its interpretation are welcome additions to an important question regarding the iron cycle in the ocean, and in principle merit the publication in a high-profile journal such as *Nature Communications*.

Besides several minor points listed below, I have, however, one main point of criticism of the present manuscript, and can therefore not recommend publication before major revision.

The manuscript refers mainly to Whitby et al, 2020, another paper on the role of humics in the global iron cycle, to show the novelty of the results and interpretations here, with respect to previous ideas. In doing so, however, it builds a strawman, and strongly misrepresents what has been said in that paper.

This already starts in the abstract that states:

"Based on iron stoichiometry with electroactive humics, it has been proposed that humic substances of terrestrial origin may form the majority of iron-binding ligands in the ocean(6). Here, we (...) demonstrate that terrigenous humics cannot account for the bulk of in-situ ligands."

Here (6) is a reference to Whitby et al, 2020. The statement that humic substances of terrestrial origin may form the majority of iron binding ligands in the ocean is never made in that paper. Instead, allochthonous humics (including riverine terrestrial as well as dust-derived) is discussed as just one component of humics, besides marine humics, derived mostly from phytoplankton. It is stated that allochthonous humics have a higher aromaticity as opposed to marine humics, which are more carboxylic-rich and/or aliphatic, consistent with the present manuscript. And moreover, it is also said explicitly that

"In our samples from the upper 1000 m of the Northeast Pacific and in all samples from the Southern Ocean, where most humic-like material comes from the recent microbial degradation of sinking autochthonous organic matter (30), the DFe/eHS ratio was below or equivalent to the lowest reported binding capacity of terrestrial humics".

So the finding that most humics in the surface Southern Ocean is not terrestrially-derived has already been made in Whitby et al.

Further down, Whitby et al. state

"When considering that a large fraction of organic matter in the deep ocean is aged and allochthonous (10), our results suggest that the persistence of terrestrial humics in the deep ocean may be pivotal in the global ocean iron cycle. However, there are many sources of both iron and humic material to seawater and it is likely that aged humics in the deep ocean are derived from a combination of terrestrial(34), marine(37) and even hydrothermal sources (16)."

It is also stated that this statement is largely based on the data from the North Atlantic and that there is a lack of deep samples in the Pacific and Southern Ocean. All in all a much more nuanced statement than the statements ascribed to it in this manuscript.

The current manuscript also states, referring to Whitby et al. that

"the study considered only terrestrially derived humics (Suwanee River fulvics and humics, further referred as HA and FA) as the main contributors to electrochemically detected iron-bound HS (eHS), despite other sources like atmospheric dust deposition (15) and microbial reworking of marine DOM (16-18) having been reported".

Besides forgetting photodegradation of terrestrial humics as another source of aliphatic DOM, this is simply a wrong representation of the statements in Whitby et al., as shown with the citations above. But it maybe explains where the authors of the present manuscript have their misconception from: Indeed the extent of iron complexation from eHS is estimated in Whitby using the published values for iron complexation for HA and Fa as bounds (which, by the way is also what is done in this manuscript. But, Whitby justify this by stating

"we account for the full range of published iron binding capacities for terrestrial standards from the Suwanee River, using an envelope that encompasses the maximum (Suwanee River Humic Acid, SRHA) and minimum (Suwanee River Fulvic Acid, SRFA) reported values (13,21,27) as a first approximation (Fig. 2). While marine humics could well incorporate a wider range, the mean iron-binding capacity of our samples (Supplementary Table S1), as well as in the Pacific Ocean (13) and Mediterranean Sea (12), indicate a similar range to terrestrial standards, lending confidence to this assumption"

This misreading of the published reference literature has to be corrected before this manuscript could be published.

Major comments

Line 25: "Thus autochthonous and freshly produced organic matter": I understand the argument for autochthonous, but not for "freshly produced". As far as I know some marine humics can also be quite refractory (CRAM), so I do not see the two as equivalent. What is the argument here?

Lines 64ff: mathematically, I do not agree that a strong positive correlation suggests that eHS represent most of the FeL pool. A positive correlation per se just emphasizes a linear relation between the two variables, but does not exclude a large y-axis intercept of the relation and also not a 1:1 slope. I assume that this statement is probably based on more than the correlation alone, but that should be stated, or else the inference be formulated more cautiously here.

Line 68-69: It is a misrepresentation that the assertion in Whitby et al, (2020) was founded on correlation. The term, used in Whitby is much more cautious: "We find broad correspondence between dissolved iron and humic ligands", immediately followed by a statement that a variability of binding capacity hinders straightforward comparisons. The main argument here is that the calculated iron-binding capacity of eHS is generally larger than the dFe concentration, with clear and systematic differences between surface/deep waters and between Atlantic/Pacific/Southern Oceans. Actually, the statement in Whitby that "In 49% of the samples shallower than 200m DFe/eHS values near or below the lower binding capacity suggested undersaturation of humics with iron", which also includes SO data, agrees well with the data shown in Figure 2b of the present manuscript.

Line 140: what is "binding complexity"? I suppose this means complexing capacity, but this remains unclear. Also, how is it calculated and does the quantity have a unit?

In lines 187 ff, Hassler estimate a contribution of dust deposition to in-situ eHS, coming to the conclusion that it is a negligible factor. While I agree to the result, the calculation done here is simply wrong: Firstly, the authors convert the deposition to a concentration by arbitrarily setting the depth of the surface layer to one metre. As the mixed layer of the ocean is typically well-mixed on timescales longer than a day, a more appropriate depth range would be the mixed layer depth, further decreasing the contribution. Secondly, a flux has the unit of a mass/m²/day; the time unit has been completely forgotten here. Dividing a flux by a depth does not give a concentration, but an increase rate of a concentration per day, or whatever time unit the flux is defined in.

Line 199-200 "that become relevant at different dFe concentrations": I would add "and at different pH values" to that, see Lodeiro, P., Rey-Castro, C., David, C., Humphreys, M.P., Gledhill, M., 2023. Proton Binding Characteristics of Dissolved Organic Matter Extracted from the North Atlantic. Environ. Sci. Technol. <https://doi.org/10.1021/acs.est.3c01810>

Line 204: i would add "surface" before SO, as this is what this paper shows. It is probably different in the deep ocean, as discussed both by Whitby et al (2020), and this manuscript. Of course one could argue that bioavailability is relevant only in the surface ocean, and this is what is talked about here. That's o.k., but maybe one should then explicitly state "phytoplankton bioavailability" as bacterial bioavailability might be different.

Figure 4B: I am missing the exact definition and unit of the quantity shown here. Also there is no explanation what the underlying red bars are. One can assume that it is the measured values, but this not stated explicitly here, and the range shown here is also not explained.

Table 1: I must admit that I failed to comprehend what is shown here, and the explanation 158-160 didn't help me really. What is the measured quantity, what is calculated from it?

Minor comments & typos

Line 59: "contrastingregions" -> contrasting regions

Line 129ff: The sentence starting with "Whereas" is incomplete, I suspect some part has been lost in a previous copyediting operation. There is also no whitespace between this and the preceding sentence.

Line 380: "measure" -> measured

Line 474: "similar .. than" does not sound right.

Line 474ff: The sentence starting with "Because" is missing a verb, I believe

Line 477: "apotential" -> a potential

Version 1:

Reviewer comments:

Reviewer #1

(Remarks to the Author)

Most of the reviewer comments have been addressed appropriately and the manuscript is far more inclusive and transparent than the previous version. There are some issues with figures, particularly fig 2, noted below. Some parts of the new manuscript do feel rushed with typos or referencing mistakes which are highlighted below, or text that assumes readers already have a greater understanding of ligands, humics etc than I would expect the general audience of this journal to be expected to have, and so it does still require some minor attention. There is still some reliance on correlations which was noted by multiple reviewers, although the text has been modified to tone down the language and a suitable rebuttal given.

There is however still an issue with the way the novelty of this study is presented. For example, in lines 188-190 at the start of the Discussion – it's great that there is now more data to further understand Fe biogeochemistry in the Southern Ocean, and this data deserves publication for that alone, as well as the expansion to including Carb and EPS, but as was pointed out by two different reviewers, the differing role of eHS in the Southern Ocean compared to elsewhere was already noted in Whitby et al. 2020 (see comments from R1 and R3). Furthermore, the difference in eHS contribution to FeL between the South Pacific, Southern Ocean and elsewhere was also recently highlighted in Whitby et al., 2024. It's fine to keep the text as it is, but it would be appropriate to reference the earlier studies here to demonstrate they are being built on and this new data augments and concurs with those earlier findings. The novelty is in the expansion of this complicated story with the inclusion of additional complementary measurements, and this still needs to be better highlighted.

Detailed comments:

The abstract is much improved, but it could still be clearer. Essentially, I think you want to say that you find that humic substances are an important part of the ligand pool for Fe in the Southern Ocean as has been found elsewhere, but that it is mostly produced in situ and composed of EPS, in contrast to other regions where terrestrial-derived humics may play a greater role(?). As written this doesn't really come across.

Figure 1B – the arrow is hiding the label of eHS, move the label so that it can be clearly seen

Figure 2:

- caption – is DDFe a typo? Also, what about FeL in this description.
- missing references within the caption text (just says 'refs' in multiple places).
- The data for the Southwest Pacific is missing from the Fig 2a, or the description for 2b should clearly state that this figure also includes data for the Southwest Pacific as well as the Southern Ocean, otherwise this is misleading.
- I cannot see lines on the figures for 'Calculated DFe saturation of eHS is shown (dotted lines) considering HA (top) and FA (bottom)' on the pdf although they are weakly visible in the tracked changes document.
- The caption states 'Relationship between eHS and L is shown for (c) different ocean basins...' the axis labels instead show DFe vs L, not eHS. Either axes or caption incorrect. Again, d says 'for Southern Ocean' when actually it is Southern Ocean + Southwest Pacific.
- L517 – typo in ligands
- For the Mediterranean Sea, only one reference is provided to Dulaquais et al. 2018, however that paper uses FeL data first published by Gerringa et al. 2017. If the Gerringa data is used it should be correctly cited. There is no Northeast Pacific FeL data in ref 14 and no other reference is provided so I'm not sure what data figure 2c and d is using for the Northeast Pacific.
- Is it necessary to distinguish between North Atlantic and Northwest Atlantic in the figure? In fact, shouldn't it be Northeast Atlantic, not Northwest, if it's referring to data from Dulaquais et al 2018? These confusions could be resolved by making it clearer to which references the data on the figure refer to. One solution would be to include the numbered references next to the basin name within the legend on each subplot, because it's difficult to associate the data on the figure with the correct references in the caption.

Figure 3 – 'as in ref 6' – ref 6 is Boyd et al., 2010? I don't think this is the correct reference. I cannot see the dotted line that represents the 100% contribution to FeL on the pdf (although I can see it on the tracked changes document). You may wish

to just list the ocean basins along the y axis instead of numbers 1 – 6 with numbered references, it would be easier to follow. Also why does the x axis on D not start at 0? Finally, similar issues to Fig 2 for references.

Figure 4 – I don't fully understand this figure, the caption could better explain what is being shown, what is meant by 'role on in situ Fe biogeochemistry'. What is 'Maximal Fe-L'?

Specific comments:

L35 – thee to the

L70 – 'despite recent data suggest' is missing something, e.g. 'which suggests' or 'suggesting'

L74 – what is the 'Uniqueness of iron..?'

L76 – move comma after life

L82-88 – give some more information on binding capacities, what it means to be similar to Suwannee river HS etc. The average reader without knowledge of humics will not understand this section.

L92 – is DFe supposed to be here or a typo

L89-92 – It could just show that humics are less saturated with DFe, which is not surprising in an Fe-limited ocean.

L89-100 – overall this section would be challenging for many readers to follow if they are not already experts in ligands and humics. I suggest making this a little more accessible to non-experts.

L108 – "first served"

L116 – what is meant by 'the bulk of eHS... cannot be described by standard FA and HA substances' – are you inferring that these eHS have a different binding capacity due to being marine-derived? Or that other ligands are important in these regions?

L121-122 – I do not understand this sentence (weaker relations to what?), plus it has multiple typos - "Coexistence of numerous compounds [change to compounds] typically results in "weaker" relations that [change to than] expected on known processes²⁸"

L124 – 'to specific microorganisms' – I did not see any data presented here on comparing to individual microorganisms, only to bacteria vs phytoplankton. Reword this for clarity.

L140 – add here "EPS, which compose part of the eHS pool, ..." (or however you wish to associate them) for clarity

L174 – add space before 'We'

L240 – add stop after inputs.

L268 – the lack of 'a' marine humic standard

L277 – add space between 'they modulate'

L288 – should this be particles

Some comments from previous review round have no response, e.g. R1 asks 'only 30s purge?' with no response, and 'A single ligand class was obtained' doesn't really address the question as to why only one ligand class was obtained when usually SA gives 2. Was an attempt made to fit multiple and it was not possible? Why might this be the case as this is unusual. I don't consider receiving responses to these queries should be an impediment to publication, but if further review iterations occur, it would be good to see the explanation.

Reviewer #2

(Remarks to the Author)

After reviewing the revised manuscript, I am pleased to confirm that the authors have significantly improved the paper. Many concerns raised in earlier reviews, including mine, have been addressed thoughtfully, resulting in a more cohesive and compelling narrative.

The flow of the manuscript has greatly improved, and even sections that I initially found problematic—such as the statistical analysis—now make more sense within the revised structure of the text. By integrating various components effectively, the authors have summarized their findings in a way that better conveys the significance and novelty of the study.

The methodological clarifications and the inclusion of additional references have strengthened the paper's foundation. The authors successfully addressed concerns about previous misinterpretations of data and justified their analytical approaches with clearer explanations. The revised figures and improved statistical presentation enhance the paper's readability and reliability.

In conclusion, this paper presents a valuable and well-substantiated contribution to the field, and I recommend it for publication.

Reviewer #3

(Remarks to the Author)

Review of the first revised version of "Marine biogenic humic substances control iron biogeochemistry across the Southern Ocean" by Hassler et al, submitted to Nature Communications

In their revised version, the authors have corrected my main concern with the first version of the paper, which was the misrepresentation of the conclusions drawn in the study by Whitby et al., 2020. They

have also clarified many of their unclear statements and now discuss much better how the different, often operationally defined, subclasses of marine DOM (humic substances, electroactive humics, ligands, EPS) relate to each other. Although the methods used do not allow to directly quantify the role of EPS in binding iron, the authors now quite convincingly show that EPS has the possibility to be an important contributor to Fe-binding ligands in the Southern Ocean and South Pacific. I would therefore now recommend publication of the study in Nature communications, after some minor revisions.

Minor comments

Line 50: 'Iron bioavailability varies ...'; the authors of course know very well that the bioavailability, as it is defined in citations 12 and 13 is not a property of seawater iron chemistry alone, but is also influenced by the uptake mechanisms of the phytoplankton species involved; maybe it would be helpful for the reader if this was mentioned briefly in this paragraph.

Line 82 ff: I find the statement made here (especially 'suggest' and 'most') maybe a bit too strong for the fact that a) the intercept 1.3 nmol/L ligand for zero eHS is not negligible, and b) that the calculation has been done using the lab-based maximum Fe binding capacity for Suwanee river humics, i.e. neglecting the competition of other cations for binding.

Besides, I do not understand to which graph the mentioned intercept and slope in line 85 belong, to Figure 2d? Maybe it would help here whether the nmol/L are actually nmol/L of Fe or of 'Fe-L'?

And, to add to my confusion: If the slope is 0.045 nmol/L / (ug eHS) (which, I assume should be 0.045 nmol/L / (ug eHS / L)), why is then 1 ug eHS equivalent to the inverse $1/0.045 = 22.2$? This whole calculation is unclear.

Line 134 ff, '...Fe bioavailability, more than Fe concentration, is key to phytoplankton photosynthesis and growth': I have some difficulty with this statement. The positive correlations of Fv/Fm with Fe-L and eHS indeed suggest that iron that is bound to ligands and electroactive humics is bioavailable. But then why is there no correlation observed to DFe? Does this suggest that some part of the DFe is NOT bioavailable, and which part would that be, given that ligand concentrations shown in Fig 2d are fairly high, and ligands are mostly unsaturated with Fe? Colloidal DFe? Iron bound to some strong ligands that are not detected?

Line 223: Given that it is based to a large part on assumed iron-binding capacities for humics, the statement 'Our study identifies EPS as the bulk of Fe-binding marine humics and ligands' is maybe a bit strong.

Line 227: The citation 56, given as support for the association of hydrothermally derived iron with (e)HS, is rather indirect: It deals with humics in hydrothermally influenced sediments. Is there more direct evidence for the association of humics with Fe in hydrothermal plumes?

Figure 1B: would it be possible to separate the labels eHS and L in the figure somewhat? And why is 'Fe-L' called 'L' in the figure and caption? In the caption it is mentioned that logK and Carb are not statistically represented. Does that mean that only a low part of their variance is explained by the first two EOFs? Then I would remove them from the plot altogether, right?

Figure 2: The dotted lines are not visible in my printout; they are visible in the pdf at some magnification. Maybe make them somewhat more prominent? In the caption, line 512: 'DDFe' -> 'DFe', Maybe ligands should also be mentioned in the bold part of the caption?

Obviously some references were intended to be included but weren't '(refs)'. What does the 'mostly this study' mean?

Line 558 'following GEOTRACES guidelines': This merits a citation.

Line 634: How large, approximately, is the error in the calculated complexation capacity, as calculated from the errors in L_{insitu} , D_{Fe} and the conditional stability constants?

Typos and small comments

Line 35: thee -> the

Line 48 and several following: I find it somewhat unfortunate to abbreviate iron-binding ligand concentrations as Fe-L, as this might be mistaken for the concentration of iron bound to ligands. Maybe L_{Fe} (possibly with Fe as subscript) is less confusing and more in line with other papers on the subject?

Line 92: I think 'South Pacific) D_{Fe} ' should just be 'South Pacific'.

Line 121: 'that' -> 'than', 'on' -> 'from'

Line 138: Here ligands are suddenly abbreviated as 'L', not as 'Fe-L'. Why?

Line 189: 'humic' -> 'humics'

Line 517: 'liagnds' -> 'ligands'. There should be a '1,' before 'Arctic'.

Line 650: '_H. Forrer' -> 'H. Forrer'

POINT-BY-POINT RESPONSE TO REVIEWERS

We are very grateful for reviewers' comments, which have greatly strengthened our manuscript. Our revised manuscript addresses all the comments and concerns listed. To show how the reviewers' comments have been addressed in our revised version of the manuscript, we copied their original reviews and added our reply below each comment in italics, with references to line numbers of the revised manuscript. Changes can also easily be assessed in the version with track changes shown.

REVIEWER COMMENTS

Reviewer #1 (Remarks to the Author):

Also included as Word doc attachment

This paper presents important new data on the ligands for iron from the Southern Ocean, an under-sampled region, and provides insight into identifying those ligands.

Strengths:

The strength in this paper lies in new data for ligands, humic substances and EPS from the Southern Ocean, an iron-limited region that is under-sampled and poorly understood. This is an important topic that deserves attention. The additional analyses into identifying the sources of the ligands are an important contribution. This study builds on previous work by the lead author into the importance of EPS as a complexing agent for iron in the Southern Ocean, and points to a biogenic source of ligands. The general assumption within the scientific community would be that most of the ligand pool in the upper Southern Ocean is biogenic, and therefore these findings are not necessarily a surprise and thus the narrative is a little off the mark, although that shouldn't discredit the importance of this new work in providing better insight into ligand sources and identity.

Text has been revised to acknowledge further studies and the fact that most of the humics are biogenic in nature. Our findings have been more clearly formulated in that we now identify the sources of key marine biogenic humics and highlight important roles for phytoplankton and bacterial EPS that are relevant to both Fe and C cycling in the Southern Ocean. Furthermore, by comparing our Southern Ocean data with data from the global ocean, we more clearly demonstrate the unique behavior or nature of eHS and L in the Southern Ocean and South Pacific.

Concerns:

This paper struggles with some key concepts and is weak on getting across its true importance, which is crucial for an article in this journal. It lacks discussion of many other key works on this topic, which for some reason are omitted from the 'global ocean' data in figure 3, yet which includes data from less relevant studies. For

example, there is no discussion of a study that already reported biologically mediated production was the main source of eHS in the Mediterranean, a nearly landlocked basin and therefore arguably more surprising to not be dominated by terrestrial humics (Dulaquais et al., 2018), or of work in the Arctic as a contrasting polar region (Slagter et al., 2019; Slagter et al., 2017). Instead, the justification for this work appears to be that the findings here are unexpected when considering a single previous study by Whitby et al. (2020), yet the conclusions of that study appear to have been misrepresented in order to fit the narrative here.

The previous work pointed out by the reviewer is now discussed and incorporated into Figs. 2 and 3 as well as Supplementary Fig. 3-4. Misinterpretation of the data presented by Whitby et al has been corrected.

The data using TAC as an exchange ligand were not considered here as TAC fails to consider all humics. That said, given the polyfunctional nature of eHS, the consideration of several exchange ligands (e.g. TAC and SA) might be beneficial as they would extend the Fe-L analytical window of detection. Such an approach would, however, require careful consideration of data detection and overlap for poorly defined eHS that is beyond the scope of this study. We nonetheless now show the Mediterranean Sea data (Dulaquais et al., 2018; although acquired with TAC) as this is the only dataset available on the role of humics in a landlocked sea. We have taken care to make clear in the relevant figure captions that this dataset was obtained using TAC.

This paper needs to be written more accurately and transparently to best place these findings in the context of existing research. The paper should be revised to better emphasize that its novelty is in answering questions posed by previous studies to enhance our understanding of this little understood topic, and in studying this important region.

We hope that our revised text is now clearer, and our findings are more deeply embedded in the context of published research. The novelty of our study is now also more clearly put forward.

The paper would also benefit from more clarity for the non-expert on what the different ligand types are, and how they are related, particularly an explanation of the crossover between EPS and humics, as well as more detail in the methods section and a more honest view of any assumptions and limitations.

We clearly define humics at the beginning of the text and more comprehensively explain that EPS are part of these humics as attested by their eHS signature (Table 1, I-150-151). More details on our calculations (Data analysis section in the methods) and related assumptions have been added (I.101-109). We also provide a short description of the sources of EPS that are relevant to the Southern Ocean, and reference relevant publications where details on their isolation and crude composition

can be found (see dedicated section in the methods).

Background info

Firstly, relevant for lines 50-55, 196 and elsewhere:

'Humic substances' is a catch-all term which represents a continuum of organic compounds following degradation of (terrestrial or aquatic) biomass, including the degradation of polysaccharides, proteins, lipids, nucleic acids, lignin, cutin, melanin etc. Within humic substances, humic and fulvic acids are merely operationally-defined fractions, with humic acids soluble in natural and alkaline solutions only, and fulvic acids soluble at all pH levels. EPS is not a separate operationally defined fraction of humic substances (as inferred in line 196).

The proposed definition of humics has been added to the text (l.57-62). EPS is now defined as a compound that is part of the marine humics pool (l. 150-151).

By definition, all soluble humic substances are a mixture of both HA-like and FA-like material; i.e. all samples contain a mixture of some ratio of material that either is soluble at all pH, or is not. HA and FA cannot empirically be distinguished from one another using the voltametric method referenced. eHS is itself an operationally defined fraction (electroactive humic-like material), and thus also a mixture of compounds that are either soluble at all pH (FA-like) or are not (HA-like). Any EPS that is detected by the eHS method would fall into either the HA-like or FA-like category purely because of this operational definition, as will many other compounds, not be distinct from it - it cannot be soluble and not be one of these if it contributes to the humic pool. The fact that some natural EPS in a sample behaves electrochemically similarly to the added FA used for the standard addition and is thus caught up in the measurement demonstrates that it shares FA-(or HA)-like properties. One can either argue that humic substances do not include EPS and thus this is an interference, or (some) EPS are HA/FA, but both cannot be true, as the authors appear to suggest.

We thank the reviewer for their input. We certainly agree with these comments and EPS is now considered as a integral contributor to the biogenic humics pool (l. 150-151.).

Finally, the eHS method cannot distinguish between sources of eHS, and cannot alone inform of the presence terrestrial or marine material, regardless of the standard used for quantification.

Yes, we do again fully agree. This limitation has been considered in our revision (l.138-139).

Literature:

There are many other relevant works that the authors may wish to consider for a

broader understanding of this topic and in building their narrative, and the recommended expanded global eHS dataset for Figure 3; it is not necessary for all of these to be included in this article (not all have the relevant data for the figure), but I share in case they are of relevance (Dulaquais et al., 2023; Dulaquais et al., 2018; Fourrier et al., 2022; Laglera et al., 2019; Mahieu et al., 2024; Muller, 2018; Slagter et al., 2019; Slagter et al., 2017; Sukekava et al., 2018; Sukekava et al., 2024; Whitby et al., 2020a; Yamashita and Nishioka, 2023; Yamashita et al., 2020).

We thank the reviewer for pointing us to these additional studies. Most of them are now included as references in the text. Three of them that used SA and measured eHS are now included in our Fig 2 and 3 to provide a better representation of the global ocean and more accurately show the uniqueness on the Southern and South Pacific Oceans. We have also chosen to show the Mediterranean Sea data (even though the L data were acquired with TAC), as this dataset contains the only measurements of humics in a landlocked sea. We have taken care to make clear in the relevant figure captions that this dataset was obtained using TAC.

Comments

Title is appropriate – *We kept it as is*

L17-18 This would benefit with a sentence introducing what humic substances are before discussing their relationship with iron.

Because humic substances are not a single component that can be shortly introduced, we decided to define them a bit further down in the introduction (l.57-62) so as to not break the flow of this first paragraph. “Humic substances” is also not a very specialized term that deserves immediate definition, although we agree that it is a central concept here.

L18 – this statement as written is incorrect, or the citation used does not support the statement.

*(Note that reference 6 in the bibliography also has a typo: Whitby, H. et al. humic-like 'substances') – *The typo has been corrected.*

The Whitby et al paper does not propose that terrestrial humics dominate the ligand pool globally. There is a suggestion that the North Atlantic specifically has a higher degree of terrestrial influence, but not that terrestrial humics form the majority of iron-binding ligands in general. There is nothing to suggest terrestrial humics would be expected to dominate the ligand pool in the surface of the Southern Ocean, and I don't believe the community think this is the case.

See figure 4 in Whitby et al which specifically separates iron limited from terrestrially-

influenced regions, and the following relevant excerpts from the Whitby et al. study:

‘Higher concentrations at low salinities demonstrate a terrestrial source, as would be expected. However, elevated concentrations in mesopelagic waters and relatively consistent values at open ocean locations also agree with a marine source and a refractory component [of humic substances]’.

‘When considering that a large fraction of organic matter in the deep ocean is aged and allochthonous¹⁰, our results suggest that the persistence of terrestrial humics in the deep ocean may be pivotal in the global ocean iron cycle. However, there are many sources of both iron and humic material to seawater and it is likely that aged humics in the deep ocean are derived from a combination of terrestrial³⁴, marine³⁷ and even hydrothermal sources¹⁶. Regardless of origin...’

‘Thus unsurprisingly, the humic contribution to iron complexation is dependent on the concentration of other, stronger ligand groups, varying regionally and with depth.’

‘Deep waters reflect an accumulation of aged humic material of higher aromaticity^{28,29}, linked to the persistence of terrestrial humics^{32,34}, the selective decomposition of organic matter³⁵, and the hypothesized aging process of humification^{24,28,34}.’

Considering all of that, I think it necessary to reconsider the justification for this study and the text alluding to this within the abstract and other parts of the paper.

All text references to Whitby paper have been corrected and our study is now more clearly justified (please also refer to other replies). Reference to many previous work using either fluorescence or electrochemical measurements to show the importance of both marine and terrestrial humics is now made in the text (e.g. l.63-69, l. 187-188).

E.g. the authors may consider something more transparent, along the lines of: Based on iron stoichiometry with electroactive humic substances, it has been proposed that humic material composes an important fraction of the iron-binding ligands in the ocean[6]. Humic substances are [definition]....., originating from a variety of sources. Some regions have a strong terrestrial humic influence, such as the Arctic Ocean[refs]. However, there is very limited data on humic ligands in the Southern Ocean, which is characterized by.....

We have considered the proposed line of argument above in our revisions at l.54-62.”

L20-22 Humics do not account for most of the ligands, but EPS account for most of

the humics and most of the ligands? Is this because only a small fraction of EPS is picked up by the humic measurements? Please clarify the wording here.

The wording has been clarified in the abstract, as follows: " Organic ligands are critical to maintaining iron in solution, but their nature is largely unknown. Coupling the iron stoichiometry of land borne humics with the marine concentrations of electroactive humic substances has shown that terrestrial organics cannot account for the majority of iron-binding ligands in the ocean. Here, we use a comprehensive dataset of electroactive humics and iron-binding ligands in contrasting regions across the Southern Ocean to show that exopolymeric substances from phytoplankton and bacteria constitute a large proportion of iron-binding substances."

The debate about how much of the EPS (mass based) contribute to eHS would break the flow and disrupt reader, nonetheless using data from Table 1, 1mg EPS contribute to 1-10% of eHS as compared with SRHA (the humics showing the strongest electrochemical signal per mg). Unfortunately, HS, eHS, EPS are all operationally defined, complex and loosely characterized materials that overlaps. We now state that EPS contribute to the eHS detected (l150-151).

L43 – how does this work fit in with the recent study (in Nature) that claims that the ligand pool is less important than previously thought and that most iron in seawater may instead exist in authigenic colloids? (Tagliabue et al., 2023).

It fits in very well as the Tagliabue et al. study showed that, in the Southern Ocean, it is organic ligands and biological uptake that determine the fate (and recycling) of Fe in surface waters. We use the fact that the Southern and Pacific Oceans, yet poorly sampled for both eHS and Fe-L, might behave differently to justify our study (l.69-72). Here we show that EPS are marine humics acting as a ligand that is produced and transformed in situ; thus, EPS are central for the biological control of Fe in surface waters. In addition, our compilation of global ocean data shows that the Southern and South Pacific Oceans behave differently from the other ocean basins and that these are the two regions controlled by ligands and biological uptake as per Tagliabue et al. Finally, we state that EPS can aggregate and form TEP, thereby "trapping" or adsorbing Fe oxide colloids (l.286-292). Hence, EPS can also be relevant not only as a marine humic able to bind Fe. While worth mentioning, we kept this latter argument succinct as it is out of the scope of our study.

L50-55 states:

'It has recently been proposed that Fe biogeochemistry across the global ocean is regulated by humic substances (HS, resulting from the degradation/humification of bacterial, plant and animal matter in soil, sediments and natural waters)⁶ . However, the study considered only terrestrially derived humics (Suwanee River fulvics and humics, further referred as HA and FA) as the main contributors to electrochemically detected iron-bound HS (eHS), despite other sources like atmospheric dust deposition¹⁵ and microbial reworking of marine DOM¹⁶⁻¹⁸ having been reported.'

The eHS method relies on a standard for quantification of eHS concentration in seawater; typically, Suwannee River Fulvic Acid or Humic Acid isolates are used, which are commercially available from the International Humic Substances Society. While a marine standard (and improved extraction procedure) would be preferable, currently there is not one commercially available at scale and the IHSS standard allows different studies to be compared with one another (as the authors do here). Terrestrial and marine humics share many similar components, and a terrestrial standard does allow quantification of the natural eHS, which in seawater is likely composed of some ratio of terrestrial and marine-derived humic material ('humic' encompassing many different things). It does not infer through its use that all of the material is of terrestrial origin. Furthermore, all humics and thus we can only assume all eHS is a variable mixture of FA-like and HA-like material, which cannot be determined using the existing method.

We agree and such shortcuts are now removed from our revised version for the sake of clarity.

The cited study uses these known operational definitions of HA and FA to explain apparent trends regionally and with depth, and to infer the likely source of the material, such as using aromaticity from fluorescence measurements to infer changes to binding capacity, based on what is known for the FA and HA standards. However, I don't believe the cited paper suggests that the FA and HA like components exclude EPS (or any of the many other components of humic substances). As stated above, EPS is not an accepted operationally defined fraction separate to FA or HA. Neither the paper cited (Whitby et al. 2020) nor this paper can empirically distinguish how much of the eHS is terrestrial, or how much actually is FA-like or HA-like. Just because an FA standard is used does not mean all of the eHS is FA, and vice versa, as the natural eHS peak can be quantified with either standard. Furthermore, as noted above the cited study discusses a variety of sources of humics, not only terrestrial, and thus the discussion is again incorrect.

We have revised the text to make it clear that given the strong relationship between in situ Fe-L and eHS and the wide agreement that humics (including terrestrial and marine humics) are important for Fe binding in most marine systems, the iron binding stoichiometry of the terrestrial standard might not be accurate and other humic compounds such as EPS can indeed represent the bulk of the Fe-L. Our study identifies compounds that prevail in the biogenic pool of humics in the Southern Ocean and likely in other places. Our observations thus provide insights into the processes behind humics and Fe cycling and highlight the need to potentially revisit the use of terrestrial standard in the future. Our study goes further in differentiating the roles for phytoplankton and bacterial EPS and clearly outlines the uniqueness of the Southern Ocean (and South Pacific) when it comes to DFe and eHS, as well as to DFe to L relationships.

L59 – add space contrasting regions- *amended*

L68-69 please clarify what is meant by ‘how the correlation between DFe and eHS that served as a foundation for this assertion is not seen (Figure 2)’. Figure 2A appears to show data from citation 6, where the data from the Southern Ocean also appear to be in a similar position to the data here in panel B – i.e. low DFe, high eHS concentrations, unless I am missing something? (noting that Figure 2A is challenging to read when using triangles for every basin, even if they are different orientations. I recommend improving the figure by changing the icons). Is the point being made here that there are some values where DFe exceeds the humic binding capacity, suggesting that other ligands are important? If so, this is interesting and needs to be stated more clearly, although if that is the intention, it does not fit with the next two lines of text that suggest that DFe concentrations were low but eHS was similar to concentrations found elsewhere (lines 71-72). Clarity needed for this section.

We have improved the visual aspect of Figure 2 as suggested, by using different shaped symbols for different ocean regions. We have also included the correlation coefficients and other statistics to underling the absence of correlation in the Southern and South Pacific Oceans, as well as the different behavior in the different ocean basins in SFig. 3.

L77 – why Liverpool Bay, which is coastal, and the Western Antarctic study, which does not have ligand data, yet many other more relevant studies are omitted? There needs to be justification for the data used in this figure – and similarly, why other studies are ignored. I strongly urge the authors to include a wider range of published humic studies in order to have a transparent and comprehensive comparison that actually is an update on the original paper. See comment under Figure 3 below.

Data from numerous additional studies have been included in Fig. 3 (the data from Liverpool Bay have been removed) to make it more robust and to better represent the existing state of knowledge on the contribution of humic substances to in situ iron-binding ligands. Incidentally, compiling these data showed a different behavior in the relationships between DFe, eHS and L in the Southern and South Pacific Oceans than elsewhere. This finding is now made clear in the text (l.89-100, 113-119), in Fig. 2 & 3 and in SFigures 3-4.

L99 – It should be noted that the percentage contribution of eHS to FeL here is similar to findings in the South Pacific (Cabanès et al., 2020; Dulaquais et al., 2023).

Yes indeed, and this has been added to the text (l.113-119 and shown in Fig. 3).

L118 – reworking of DOM, and also of POM - see ‘Contribution of electroactive humic substances to the iron-binding ligands released during microbial remineralization of

sinking particles' (Whitby et al., 2020a). It may be relevant elsewhere in this paper too as it includes data from the Southern Ocean.

Yes, this has been added to the text (e.g., l.65-67).

L126 – how is the amount of ligand that is allochthonous determined? (and similarly elsewhere, the amount of eHS that is terrestrial?) or is this purely through order of elimination when using measurements of known biogenic components? Clarity needed here.

We agree that neither eHS nor Fe-L allow for the distinction of different compounds. This is done purely by investigating various types of humics and comparing the results obtained for each of them. We hope that this point is now clearer in the revised text (see l.138-139).

L129 – add space before 'Whereas' and then either reword this sentence so it is a complete standalone sentence, or join it with the subsequent sentence '...phytoplankton, carbohydrates...' – there is also a typo there and the word should be 'constitutive'.

Changes have been made as per suggestions.

L193 – add is 'is what determines'. *Amended*

L205 – 207 reword this section to better convey the true novelty and importance. Remove revisit – see above, there is nothing to revisit, just build on. What is important here is that we have new data for the Southern Ocean, and further evidence that eHS has a strong biogenic component, supporting previous studies e.g. Dulaquais et al (2018). The authors could use this work to better clarify the relationship between EPS and HS.

We have revised our text to address the reviewer's comments and have more clearly emphasized the novelty of our study (l.21-28, l.275-275, l.293-299).

One could argue, if there is a lot of excess ligand here, and a similar amount of humic material here to other regions, it is not then purely the low iron supply that is limiting productivity, and the ligands themselves have little impact on iron geochemistry, no matter what they consist of? The message of why this study is important does not come across as powerfully as it should for a Nature Communications article.

The ligands themselves will still play a role in maintaining Fe bioavailability and enhancing Fe recycling and exchange between ligands and cells in surface water. Weaker ligands, such as EPS, need to be present in greater amounts to outcompete strong ligands. So, we think that the high ligand excess here might be required to support Fe recycling and maintain Fe more efficiently in surface water, also preventing Fe colloidal oxide scavenging. Anomalies seen in the Southern and South

Pacific Oceans (compared to the rest of the basins) (Fig 3, SFig. 3 &4) suggest that high ligands of biological origin have an important role in keeping Fe in surface waters in its bioavailable form. Moreover, our study shows that EPS play a key role in this. Our results also support the ligand and biological uptake control of Fe in surface water, as put forward by Tagliabue et al. (2023).

I recommend the authors try to emphasise why it is important to know that most of the eHS present is of biogenic origin and thus explain the true novelty of this study.

We demonstrated and emphasized the unique behavior of eHS, DFe and L in the Southern and Pacific Ocean). The findings that most ligands are of biological origin is relevant to Fe cycling and residence time, Fe bioavailability as stated at several places in the text. We further emphasize its significance to the biological carbon pump at the end (l.293-2999. Yet to really scale-up and investigate potential consequences of our findings, this needs to be explored using modeling, which we will for sure be very keen to contribute to. Personally,, as a chemist it suggests that labile compounds and kinetics are critical to drive the system, yet it goes out of the scope of this study and unfortunately our qualification of labile compounds is likely flawed as it does not have the time to accumulate in water. Measuring kinetics at sea under relevant conditions is not straightforward and out of the scope of the present work.

Figures

Figure 1 is clear and well explained. However, I think it is a shame not to have the map shown as extended fig 1 within the main text.

The map has been added.

Figure 2A – use different symbols instead of all circles – even in different orientations, they are still challenging to distinguish between (and maybe more so for those with visual disabilities or dyslexia for instance).

Amended.

Figure 3 - In general I like this figure, but as currently presented it is misleading as it cherry-picks only some published humic data to support a certain narrative, omitting other data, and with incorrect labels.

In panel C, 'Global Ocean' should either read 'North Atlantic Ocean', as according to the paper, only the samples where both humic and ligand data was available was used for this percentage which was only the N Atlantic, or actually be a compilation of the existing global data that is available. I strongly urge the authors to do the latter.

We have modified this figure as per recommendations. We now include a much wider range of data from the literature

For example, multiple studies have shown eHS to compose around 20%–60% of the iron-binding ligand pool (Dulaquais et al., 2018; Fourquez et al., 2023; Whitby et al., 2020b), while iron-humic complexes can be the dominant form of dissolved iron in the terrestrially-influenced Arctic Ocean (Laglera et al., 2019; Slagter et al., 2019). There is also another study in the Pacific which corroborates the Cabanes study with low eHS relative to DFe (Dulaquais et al., 2023). Combining these would give a more comprehensive global dataset. I therefore encourage the authors to either separate the data by basin, or, to have a Southern Ocean and ‘Global’ or ‘Rest of ocean’ dataset, to be truly transparent.

This has been done and now clearly reflects the unique behavior in the Southern and South Pacific Oceans.

Figure 4, Table 1, and section L135 -169 – This section and the linked visual material is challenging to follow, particularly considering line 150 ‘Together, this shows that EPS ability to bind Fe cannot be predicted from their hydrolysable carbohydrate content’. This section would be improved by a simple explanation for the lay audience (not EPS/ligand or DOM experts) explaining what is being compared and how they are related, particularly relating to the different forms of EPS and how they relate to eHS. Out of curiosity, how do EPS compare to other humic measurements, such as fluorescence measurements?

We tried to improve it and define terms up front. Yet EPS, HS and eHS are all operationally defined as well as complex and poorly characterized materials. While we know they share similarity (now stated in the text, we do not know up to where their contribution to eHS overlaps and whether it is due to similar functionality or not. We do not have yet fluorescence measurement on EPS. Unfortunately, the field is lacking data to unequivocally address this comment.

Methods

I hoped the methods section would give a more detailed explanation of some concepts that are difficult to follow in the main text, but it is rather sparse and missing some key details.

Additional details have been added, such as basic formulas for calculations and more information about the EPS described in this study.

General:

Ensure throughout that units are correctly written, e.g. μg instead of ug. *Done.* There is confusion with references – for example, on line 477 in methods, quotes reference 60, but there are only 19 references in the methods and 50 in the main text, so which is reference 60? *This was an error and it should be 19- this was corrected.*

This may be up to the journal, but with the separate bibliography and numbering system restarting in the methods, there are then some duplicate references where a

single paper has two numbers depending on the section being read, which can make it confusing for the reader and possible to accidentally read the wrong bibliography and use the wrong reference. Is it possible to have a continuation of the original numbered reference list in the methods so each paper has only one number?

Indeed this was a mistake, as ref 60 should have been ref 19. Journal format will be carefully looked at in our revisions, as will references.

It looks like there is the inclusion of some methods for data that is not shown or discussed in the main text; depending on the rules of the journal, I would recommend moving any methods relating to only supplementary figures and tables to the supplementary info so that space can be saved in the methods to better explain some of the key parameters.

This was done as suggested.

Specific: All specific comments have been considered.

L393 - missing some punctuation?

L401 – HNO₃

L420 – remove stop after ‘as per 6’

L422 – remove comma after ‘done’

L428 – only 30s purge?

L430 – add space ‘and the’

L432 – why was only one ligand class fitted when SA typically gives multiple classes? It would have been interesting to see if two ligand classes were given and to compare the log K values of L1 and L2. *A single ligand class was obtained.*

L434 – why was the Laglera and van den Berg 2009 method used and not the updated method (Sukekava et al., 2018)? What is the full code for the IHSS standard; was the FA standard saturated with Fe prior to use? How many standard additions were performed? How was the detection limit determined? All details can be found in the data reference (DL, 4 standard additions were made- me to extract info here).

The new method described in 2018 was not implemented as our analyses were done in 2017-2018. In the future we will use this revised technique although, due to low dFe in the Southern Ocean, we are not sure we would be able to detect in-situ Fe-HS in many cases.

L474 – change ‘than in’ to ‘as in’. Typo in ‘largely’ *Done*

L476-479 Wording here is unclear. Used in situ ligand data instead of DFe but for what? They are different things. Comparing to L gives an estimate of the % composition of L as eHS (ignoring that L is controlled by the artificial ligand selected, the detection window, pH buffered to etc). Comparing to DFe gives the relationship between the eHS and DFe, in turn ignoring that eHS can bind other metals and that DFe can be bound to other ligands. Both require assumptions, both are useful

comparisons that give you an insight, but insights into different things and as written this section does not make sense. It's impossible to check the reference 60 as it doesn't exist in the bibliography.

This is now clarified in the methods, and assumptions are also explicitly outlined.

L477 – space between a potential

L480-482 – again this whole section needs to be written more clearly. 'we were able to transform measured eHS in an estimated Fe binding ligands concentration' does not make sense grammatically. Also, please detail how this was done, give references/calculations/assumptions etc. *This has been added.*

L489 – 'should be equal to, or exceed, the'

L490 – 'with respect to different ligands' and 'investigation into the respective roles'

See attached for bibliography.

This bibliography has been incorporated to the text as per reviewer suggestion and the dataset associated with publication in blue are included in Fig 2 and 3. Except for the Mediterranean Sea, we have not considered the studies using TAC as an exchange ligand as this technique has several limitations related to humics as summarized in Slagter et al 2017 (Marine Chemistry, 197),.

Dulaquais, G. et al., 2023. The role of humic-type ligands in the bioavailability and stabilization of dissolved iron in the Western Tropical South Pacific Ocean. *Frontiers in Marine Science*, 10.

Dulaquais, G. et al., 2018. The biogeochemistry of electroactive humic substances and its connection to iron chemistry in the North East Atlantic and the Western Mediterranean Sea. *Journal of Geophysical Research: Oceans*.

Fourrier, P. et al., 2022. Characterization of the vertical size distribution, composition and chemical properties of dissolved organic matter in the (ultra)oligotrophic Pacific Ocean through a multi-detection approach. *Marine Chemistry*, 240: 104068.

Laglera, L.M. et al., 2019. First quantification of the controlling role of humic substances in the transport of iron across the surface of the Arctic Ocean. *Environmental Science & Technology*, 53(22): 13136-13145.

Lodeiro, P., Rey-Castro, C., David, C., Humphreys, M.P., Gledhill, M., 2023. Proton Binding Characteristics of Dissolved Organic Matter Extracted from the North Atlantic. *Environ. Sci. Technol.* <https://doi.org/10.1021/acs.est.3c01810>

Mahieu, L. et al., 2024. Iron-binding by dissolved organic matter in the Western Tropical South Pacific Ocean (GEOTRACES TONGA cruise GPpr14). *Frontiers in Marine Science*, 11: 1304118.

Muller, F.L.L., 2018. Exploring the Potential Role of Terrestrially Derived Humic Substances in the Marine Biogeochemistry of Iron. *Frontiers in Earth Science*, 6(159).

Slagter, H.A., Laglera, L.M., Sukekava, C. and Gerringa, L.J.A., 2019. Fe-binding organic ligands in the humic-rich TransPolar Drift in the surface Arctic Ocean using multiple voltammetric methods. *Journal of Geophysical Research: Oceans*, 0(ja).

Tagliabue, A. et al., 2023. Authigenic mineral phases as a driver of the upper-ocean iron cycle. *Nature*, 620(7972): 104-109. Whitby, H. et al., 2020a. Contribution of electroactive humic substances to the ironbinding ligands released during microbial remineralization of sinking particles. *Geophysical Research Letters*, 47(7): e2019GL086685.

Sukekava, C. F., Downes, J., Filella, M., Vilanova, B., & Laglera, L. M. (2024). Ligand exchange provides new insight into the role of humic substances in the marine iron cycle. *Geochimica et Cosmochimica Acta*, 366, 17-30.
<https://doi.org/10.1016/j.gca.2024.05.002>

Yamashita, Y., Nishioka, J., Obata, H. and Ogawa, H., 2020. Shelf humic substances as carriers for basin-scale iron transport in the North Pacific. *Scientific reports*, 10(1): 1-10

Reviewer #2 (Remarks to the Author):

The manuscript entitled “Marine Biogenic Humic Substances Control Iron Biogeochemistry across the Southern Ocean” presents a comprehensive investigation into humic substances, with particular emphasis on extracellular polymeric substances (EPS) as a crucial component. Although only 1% of iron appears to be complexed with these substances, the study is original and makes a significant contribution to the field by addressing a critical gap in the current literature. The importance of exopolysaccharides in the pool of iron ligands, as well as their substantial influence on the humic fraction, is well-documented. However, this paper effectively demonstrates, through a large dataset, the specific influence of these substances on iron biogeochemistry.

It is not 1% of the EPS that is occupied by Fe, as we have no means of quantifying this directly. Rather, since EPS are an iron ligand that is not made by design, it will likely react with a lot of different dissolved metals; we stipulate that with only 1% of the EPS occupied by Fe, they still play a role. Based on ICP MS measurements of isolated EPS, Fe was always the most dominant trace metal (up to 10x more abundant than Zn and 50 x more than Cu). Therefore, the 1% Fe saturation level of EPS is a reasonable assumption here. This has been made clear in the text (l.172-

174).

While the authors have successfully identified and addressed this research gap, some methodological and analytical choices require further consideration. Specifically, the use of correlations with low p-values but weak r-values suggests only a modest influence between the variables, rather than a strong correlation. Although the relationship is not random, the weak correlation raises concerns about the strength of the inferred relationships between the parameters.

The only relationship that is truly central to this study is the one between L and eHS – for this, the R value of between 0.7 and 0.8 is indeed high, indicating a strong correlation. The other (weaker) relationships are briefly discussed as they point to a biological source/sink for ligands. Since these relationships, as the reviewer concedes above, are not random, we feel justified in including them to make this point. Relationships between chemical and biological parameters that do not have the same turnover rates are always challenging to establish and weaker than expected based on known processes, particularly given that DOM is a “potpourri” of compounds with different sources and liabilities. This point was made clear in the study of Smith et al (2022) that we now cite in the text. We have thus decided to keep our references to the weaker relationships in the manuscript but to rework the text to make the implications clearer (l.121-136).

Although the data presented are robust and the experimental design is generally well-structured, the reliance on these correlations may not sufficiently support the accuracy of the data analysis. It is recommended that the authors minimize the emphasis on these correlations as currently presented.

Correlations are used to infer a biological source for, and consumption of, eHS and L in other previous studies.. Nonetheless, correlations are described in the text as “weak” and the word “underscore” was replaced by “suggest” so as not to overstate the point, as per the reviewer’s suggestion. The attenuation of hydrolysable saccharide is not linear with depth, which might explain the low R value. However, previous studies have shown similar trends, making us reluctant to change/remove the text. Overall, we feel that this section is already very short. Additionally, such large-scale dataset that can be used to explore these correlations are very limited; hence we think it is worth retaining this brief analysis.

While the methodologies employed are appropriate for addressing the research question, it is unfortunate that the authors did not utilize a more recent version of the method for humic substances, such as the updated approach described by Laglera et al. (2007). The use of a more current methodology could have provided additional insights, particularly given the reliance on eHS results. Nevertheless, this does not detract from the overall quality of the study, which remains well-structured and informative.

As noted in response to reviewer 1, the eHS analyses were done prior to the publication of this new method, which we will implement in future work. Nonetheless, our calculation approach gave good agreement with the numbers obtained by Laglera al. (2019) for the Arctic; this point is made clear in the text (l.111-113).

In conclusion, I recommend acceptance with minor revisions.

We thank reviewer 2 for their positive assessment of our work.

GENERAL OVERVIEW

The paper provides a good explanation of how ligands and humic substances interact. However, the explanation is not always clearly presented, leading to some confusion in certain sections. I strongly suggest reading papers that describe the kinetics of humic substances with other ligands, as these could help address some of the information gaps.

We have added one reference of exchange (Sukekava et al 2024, l. 270-273) and highlighted as future research, yet we did not want to go too much in this direction to keep a clear focus and a good flow in our story.

Recommendations and Questions

-Line 28: Change “de facto” to “in practice.” *We kept “de facto” as “in practice” does not read as well and this suggestion is purely stylistic.*

- Line 54: The paper should not focus exclusively on SRFA and SRHA as the only methods for measuring humic substances. This sentence appears too restrictive. The current phrasing implies that these are the only approaches, which is not accurate. Consider rewording to allow for alternative methods in the study of humic substances.

We decided to focus on eHS as they are Fe-binding humics, which are central to our study. However, we now also refer to FDOM studies in the text (l.54-57). For our approach, we needed to consider eHS, Fe-L, dFe and log K', which justifies our choice of methods.

- Line 59: Change “contrastingregions” to “contrasting regions.” *Amended*

- Line 61: Despite the significant p-value, the correlation is so small that it is difficult to believe any inverse relationship exists.

The attenuation of hydrolysable carbohydrate may not be linear with depth; a similar trend was observed in the South Pacific (Cabanes et al 2020). Yet this was observed with other markers for carbohydrates, and it was not added here to ease the flow of

reading. This statement is backed up by solid observations made using FT-ICR-MS and NRM (Lechtenfeld et al., 2014; Hetkorn et al., 2006). As such, we have maintained text as is.

- Line 65: Did you extract the percentage of eHS to represent binding capacity? Since you transformed humic substances into eHS using the binding capacity of SRFA and SRHA, these percentages could help reinforce the strong positive correlation.

We do not fully understand the point made point here. Which percentage? We did our calculation as per several published studies to which we refer in the manuscript (e.g. I. 104). We have added calculation details to the method section that should make this specific aspect clearer.

- Line 68: There are many other papers that describe this, such as Laglera et al. (2019). Be sure to reference relevant literature in the discussion, including Laglera et al. (2019), which provides additional insights into this topic.

The reference has been added together with other relevant references in the discussion – please see references added above (end of reviewer 1 reply).

- Line 69: It is unfortunate that the method was not updated, as using a more recent methodology would have increased the value of this data. By applying the updated approach, it would have been possible to observe the amount of iron complexed with humic substances and to calculate the percentage of iron bound to humic substances. Despite this limitation, I agree that deep-water data could have had a greater influence on this outcome. Please reference the relevant paper that describes this issue.

We have replied to this point above and we agree that the revised method needs to be used in the future. That said, in the Southern Ocean where DFe is very low, the new method might find eHS-bound Fe to be below the detection limit.

- Line 70: The comparison method in Figure 2b seems unnecessarily complex, despite the values being similar. I suggest changing the triangle symbols to different shapes to improve clarity. Adjusting the graphical representation may help the reader interpret the data more easily, without introducing unnecessary complexity.

This figure has been augmented with data from four additional studies to better place the Southern Ocean within the global ocean context, and the symbols have been changed to improve readability. Please also refer to our reply to reviewer 1 in this regard. The comparison with other ocean basins clearly reveals the uniqueness of the Southern and South Pacific Oceans. This is represented in Fig. 2 and SFig 3-4.

- Line 77: I disagree with the comparison of Liverpool Bay to other oceans. It would

be more appropriate to use data from regions with similar environmental conditions or from areas with well-known concentrations of humic substances, such as the Arctic Ocean. The Arctic demonstrates high concentrations of humic substances and has a significant influence on the ligand pool. Liverpool Bay, with its distinct riverine inputs from major rivers such as the Alt, Clwyd, Dee, Ribble, and Mersey, and unique circulation patterns, is less comparable to open ocean environments.

We have now included Arctic data and removed the Liverpool Bay data as recommended.

- Line 87: Change “theFe-binding” to “the Fe-binding.” *Done.*

- Line 95: This could be associated with the presence of other ligands. It is well known that if DFO binds to iron before humic substances, the latter cannot displace it. This explanation should reference Sukekava et al. (2024) and the concept of 'first come, first served.' A discussion on ligand competition, particularly about existing literature, would enhance this explanation.

This is now mentioned in the text (l.107-108).

- Line 102: The examples provided do not mention the Southern Ocean. I suggest reformulating the sentence to clarify that humic substances have been identified in various regions globally. A clearer description of the global distribution of humic substances would improve the reader's understanding.

The text has been modified to include recent works, including two from the Southern Ocean (Smith et al., 2022, Whitby et al 2020), and to provide a clear global view of humics. Only limited values from Whitby (eHS) were added to Fig. 2 and 3 (most of the data is from our study as this region has been poorly sampled for this parameter).

- Line 133: There is a need to reinforce the importance of conducting kinetic studies to address uncertainties about EPS and their classification as weak ligands. This could be an important measurement for future studies.

We fully agree that the fact that most of marine humics relevant to iron biogeochemistry are labile compounds in surface water call for further studies in this direction. As stated before, this has been added as a future research direction at the end of our manuscript (l.270-273). However, it is not extensively discussed here as it is not one the scope of this manuscript.

- Line 189: The data is particularly interesting, as no riverine input seems capable of explaining the results. Dust and autochthonous humic substances appear to be the only plausible sources.

Indeed, autochthonous humics are the key sources here, as shown in our study.

- Line 390: I assume all the materials for sampling and analysis were prepared following the GEOTRACES protocol. Should this be explicitly mentioned in the paper?

Yes, it is the case and now appears in the method section.

- Line 412: Has the zinc concentration data been utilized at any point in the analysis? If not, is it necessary to describe it in the text?

Text referring to the Zn data has been removed.

- Line 433: Why were calculations performed with only one ligand class? Could this be clarified further?

Only one ligand class has been detected.

- Line 477: The reference appears to be incorrect; I could not locate any reference numbered 60. Please verify this.

Apologies for the typo, we meant to cite reference 19, referring to Whitby et al 2020 Sci.Rep.

- Line 482: I am unclear on how you calculated the different ligand concentrations for humic substances and carbohydrates. Could you provide a more detailed explanation of this section?

Calculation details have been added in the method section. Similar calculations were done in previous studies, which are cited in the text.

- Extended data Figure 1: While visually appealing, it is not clear how this figure contributes to the overall paper. It may be more effective to display only the sample locations without including the nitrate variability.

A figure showing sampling sites was requested by reviewer 1 to be included in the main text. We think that adding nitrate provides important context since, while not directly relevant to this study, this macronutrient is relevant for co-limitation in the Sub-Antarctic Zone and might be interesting to the reader. The figure also clearly (and immediately) shows that we sampled contrasting regions of the SO. Because this information does not distract from the visualization of the stations, we have decided to keep the figure as is.

- Extended data Table 1: If all the sampling locations are already included in Figure 1, I would recommend omitting this table, as it may be redundant.

We disagree here, sampling time and lat/long information might be important to readers, for example to identify close-by dataset for further comparison. As this is online material, we have decided to keep it as is.

Extended data Figures 2 and 3, I recommend placing greater emphasis on the distinct behaviors observed across various locations, rather than focusing primarily on correlations between variables. Different regions exhibited unique patterns, particularly in ligand distribution. For instance, the High Nutrient Low Chlorophyll (HNLC) region and the Mertz Polynya displayed distinct groups of carbohydrates and eHS. These differences could be better integrated to highlight the varying chemical influences across different water masses, especially considering the influence of differing oceanic currents, as described in Smith et al. (2022) and Ardiningsih et al. (2020, 2021).

While we agree regarding the patterns, and especially in the Mertz polynya, geographic variation within the Southern Ocean is not the focus of this study. Instead, we want to emphasize a general pattern of biologically in situ produced humics that control Fe biogeochemistry across contrasting region of the Southern Ocean (top 1000 m), rather than pointing to regional differences in humics compositions. So, we have not changed the text. We do acknowledge that the reviewer's remark here is very interesting and that regional differences within the Southern Ocean are undoubtedly worth looking at in future field investigations.

Sukekava, C. F., Downes, J., Filella, M., Vilanova, B., & Laglera, L. M. (2024). Ligand exchange provides new insight into the role of humic substances in the marine iron cycle. *Geochimica et Cosmochimica Acta*, 366, 17-30. <https://doi.org/10.1016/j.gca.2024.05.002>

Laglera, L. M., Sukekava, C., Slagter, H. A., Downes, J., Aparicio-Gonzalez, A., & Gerringa, L. J. (2019). First quantification of the controlling role of humic substances in the transport of iron across the surface of the Arctic Ocean. *Environmental Science & Technology*, 53(22), 13136-13145. <https://doi.org/10.1021/acs.est.9b04208>

Ardiningsih, I., Sander, S.G., Gerringa, L.J.A., Laan, P., & Middag, R. (2020). Sources of Fe-binding organic ligands in surface waters of the western Antarctic Peninsula. *Marine Chemistry*, 226, 103855. <https://doi.org/10.1016/j.marchem.2020.103855>

Ardiningsih, I., Seyitmuhammedov, K., Sander, S.G., Stirling, C.H., Reichart, G.-J., Arrigo, K.R., Gerringa, L.J.A., & Middag, R. (2022). Fe-binding organic ligands in coastal and frontal regions of the western Antarctic Peninsula. *Frontiers in Marine Science*, 9, 883482. <https://doi.org/10.3389/fmars.2022.883482>

Smith, P., Thompson, L., Williams, J., & Baker, A.R. (2022). Identifying potential sources of iron-binding ligands in coastal Antarctic environments and the wider Southern Ocean. *Frontiers in Marine Science*, 9, 948772.

Reviewer #3 (Remarks to the Author):

Review of "Marine biogenic humic substances control iron biogeochemistry across the Southern Ocean" By Christel Hassler et al., submitted to Nature Communications, 2024

Scope of the manuscript, general assessment and recommendation

The manuscript by Hassler et al. is a contribution to the long-standing and biogeochemically very important question concerning the nature of the organic ligands that have been shown to control iron residence time and bioavailability in the ocean. It does so based on a new data set of measurements of electroactive humic substances, iron binding ligands, and dissolved iron concentrations mostly from the upper 200m depth below the surface in the Southern Ocean. It is argued that the bulk of the electroactive (iron-binding) humics and ligand iron binding in their samples is based on exopolymeric substances produced by bacteria and phytoplankton, as part of the marine humic pool.

Both the data set and its interpretation are welcome additions to an important question regarding the iron cycle in the ocean, and in principle merit the publication in a high-profile journal such as Nature Communications.

Besides several minor points listed below, I have, however, one main point of criticism of the present manuscript, and can therefore not recommend publication before major revision.

The manuscript refers mainly to Whitby et al, 2020, another paper on the role of humics in the global iron cycle, to show the novelty of the results and interpretations here, with respect to previous ideas. In doing so, however, it builds a strawman, and strongly misrepresents what has been said in that paper.

The fact that the importance of biogenic humics is widely acknowledged, including in the Whitby papers but also in other works, is now clearly put forward in our revised manuscript. Additional studies have been included, allowing us to better place our Southern Ocean data into the context of the global ocean (in figures and discussion). This avoids building the "strawman" suggested by the reviewer and also better represents the recent literature.

The novelty of our study in identifying the source (and processes) associated with these biogenic humics and demonstrating their potential in controlling Fe

biogeochemistry across contrasting regions of the Southern Ocean is now put forward more prominently. We also mention the need for new marine humic standard materials for future process studies and propose biological EPS as a relevant material as it can be easily isolated in bulk quantities. – Indeed, we intend to isolate relevant material directly at sea in the near future. Finally, the data compilation from the global ocean clearly reveals the unique features of the Southern and the South Pacific Oceans (see Fig. 2 and SFig. 3-4).

This already starts in the abstract that states:

"Based on iron stoichiometry with electroactive humics, it has been proposed that humic substances of terrestrial origin may form the majority of iron-binding ligands in the ocean(6). Here, we (...) demonstrate that terrigenous humics cannot account for the bulk of in-situ ligands."

The abstract has been fully revised to take the reviewer's recommendation in consideration.

Here (6) is a reference to Whitby et al, 2020. The statement that humic substances of terrestrial origin may form the majority of iron binding ligands in the ocean is never made in that paper. Instead, allochthonous humics (including riverine terrestrial as well as dust-derived) is discussed as just one component of humics, besides marine humics, derived mostly from phytoplankton. It is stated that allochthonous humics have a higher aromaticity as opposed to marine humics, which are more carboxylic-rich and/or aliphatic, consistent with the present manuscript. And moreover, it is also said explicitly that

"In our samples from the upper 1000 m of the Northeast Pacific and in all samples from the Southern Ocean, where most humic-like material comes from the recent microbial degradation of sinking autochthonous organic matter (30), the DFe/eHS ratio was below or equivalent to the lowest reported binding capacity of terrestrial humics".

As outlined above, this has been corrected as recommended.

So the finding that most humics in the surface Southern Ocean is not terrestrially-derived has already been made in Whitby et al.

Yes, indeed. However, the novelty of our study relates to its far larger representation of the Southern Ocean, across contrasting regions, which demonstrates that the Southern Ocean behaves differently from other ocean basins. Additionally, we identify what these marine humics are and demonstrate their role in controlling in situ Fe biogeochemistry. The differentiation between phytoplankton and bacterial EPS is also a novelty.

Further down, Whitby et al. state

"When considering that a large fraction of organic matter in the deep ocean is aged and allochthonous (10), our results suggest that the persistence of terrestrial humics in the deep ocean may be pivotal in the global ocean iron cycle. However, there are many sources of both iron and humic material to seawater and it is likely that aged humics in the deep ocean are derived from a combination of terrestrial(34), marine(37) and even hydrothermal sources (16)."

It is also stated that this statement is largely based on the data from the North Atlantic and that there is a lack of deep samples in the Pacific and Southern Ocean. All in all a much more nuanced statement than the statements ascribed to it in this manuscript.

The current manuscript also states, referring to Whitby et al. that

"the study considered only terrestrially derived humics (Suwanee River fulvics and humics, further referred as HA and FA) as the main contributors to electrochemically detected iron-bound HS (eHS), despite other sources like atmospheric dust deposition (15) and microbial reworking of marine DOM (16-18) having been reported".

The text has been modified to clearly state that terrestrial humics are the standard material used to derived LHS, while also noting that terrestrial material does not constitute the bulk of the humics reported in the ocean.

Besides forgetting photodegradation of terrestrial humics as another source of aliphatic DOM, this is simply a wrong representation of the statements in Whitby et al., as shown with the citations above. But it maybe explains where the authors of the present manuscript have their misconception from: Indeed the extent of iron complexation from eHS is estimated in Whitby using the published values for iron complexation for HA and Fa as bounds (which, by the way is also what is done in this manuscript. But, Whitby justify this by stating

"we account for the full range of published iron binding capacities for terrestrial standards from the Suwanee River, using an envelope that encompasses the maximum (Suwanee River Humic Acid, SRHA) and minimum (Suwanee River Fulvic Acid, SRFA) reported values (13,21,27) as a first approximation (Fig. 2). While marine humics could well incorporate a wider range, the mean iron-binding capacity of our samples (Supplementary Table S1), as well as in the Pacific Ocean (13) and Mediterranean Sea (12), indicate a similar range to terrestrial standards, lending confidence to this assumption"

This misreading of the published reference literature has to be corrected before this manuscript could be published.

This has now been thoroughly modified. Please also see our response to reviewer 1, who identified a number of similar issues.

Major comments

Line 25: "Thus autochthonous and freshly produced organic matter": I understand the argument for autochthonous, but not for "freshly produced". As far as I know some marine humics can also be quite refractory (CRAM), so I do not see the two as equivalent. What is the argument here?

This sentence refers to our finding on phytoplankton and bacterial EPS - which are representative of freshly produced in situ material and not CRAM. Unfortunately, we do not have deep bacterial EPS or CRAM products isolated for the deep ocean. This was already stated in the original version of the manuscript, and remains in the revised version. It is also clearly stated in the text that aged (or even terrestrially derived) material is important.

Lines 64ff: mathematically, I do not agree that a strong positive correlation suggests that eHS represent most of the FeL pool. A positive correlation per se just emphasizes a linear relation between the two variables, but does not exclude a large y-axis intercept of the relation and also not a 1:1 slope. I assume that this statement is probably based on more than the correlation alone, but that should be stated, or else the inference be formulated more cautiously here.

We agree that correlations can be misinterpreted and/or misleading, yet the positive correlation between eHS and FeL nonetheless suggests a strong link between these two parameters. Also, our statement to this effect is supported by the correlations as well as the ACP and previous literature showing a key role for humics and eHS. The intercept is at 1.3 nM L on the y axis and the slope is 0.045 nmol L/ mug eHS, so meaning a slope corresponding to 22.2 nmol L per µg eHS with is close to the average DFe/eHS observed and the values reported for standard Suwanee River humics. This consideration was now added in the text (l.84-87)

Line 68-69: It is a misrepresentation that the assertion in Whitby et al, (2020) was founded on correlation. The term, used in Whitby is much more cautious: "We find broad correspondence between dissolved iron and humic ligands", immediately followed by a statement that a variability of binding capacity hinders straightforward comparisons. The main argument here is that the calculated iron-binding capacity of

eHS is generally larger than the dFe concentration, with clear and systematic differences between surface/deep waters and between Atlantic/Pacific/Southern Oceans. Actually, the statement in Whitby that "In 49% of the samples shallower than 200m DFe/eHS values near or below the lower binding capacity suggested undersaturation of humics with iron", which also includes SO data, agrees well with the data shown in Figure 2b of the present manuscript.

This section has been reworded.

Line 140: what is "binding complexity"? I suppose this means complexing capacity, but this remains unclear. Also, how is it calculated and does the quantity have a unit?

Indeed, thank you for pointing this out. How it is calculated is shown in the text. We use L^ (L_t -dFe) multiplied by $\log K'_{FeL}$ for the in situ one; L^* is multiplied by the other $\log K$ to give the potential of other ligands to contribute here. This method has been discussed with Martha Gledhill and we acknowledge many simplifications (1:1 Fe:L and no kinetics of exchange, and no polyfunctional ligands), however these simplifications are required to test this hypothesis.*

In lines 187 ff, Hassler estimate a contribution of dust deposition to in-situ eHS, coming to the conclusion that it is a negligible factor. While I agree to the result, the calculation done here is simply wrong: Firstly, the authors convert the deposition to a concentration by arbitrarily setting the depth of the surface layer to one metre. As the mixed layer of the ocean is typically well-mixed on timescales longer than a day, a more appropriate depth range would be the mixed layer depth, further decreasing the contribution. Secondly, a flux has the unit of a mass/m²/day; the time unit has been completely forgotten here. Dividing a flux by a depth does not give a concentration, but an increase rate of a concentration per day, or whatever time unit the flux is defined in.

The text has been modified to compare daily eHS daily input from dust to mixed layer depth and associated eHS levels (l. 231-240).

Line 199-200 "that become relevant at different dFe concentrations": I would add "and at different pH values" to that, see Lodeiro, P., Rey-Castro, C., David, C., Humphreys, M.P., Gledhill, M., 2023. Proton Binding Characteristics of Dissolved Organic Matter Extracted from the North Atlantic. Environ. Sci. Technol. <https://doi.org/10.1021/acs.est.3c01810>

This has now been added with the appropriate reference (l.259).

Line 204: i would add "surface" before SO, as this is what this paper shows. It is probably different in the deep ocean, as discussed both by Whitby et al (2020`), and

this manuscript. Of course one could argue that bioavailability is relevant only in the surface ocean, and this is what is talked about here. That's o.k., but maybe one should then explicitly state "phytoplankton bioavailability" as bacterial bioavailability might be different.

Surface was added as suggested.

Figure 4B: I am missing the exact definition and unit of the quantity shown here. Also there is no explanation what the underlying red bars are. One can assume that it is the measured values, but this not stated explicitly here, and the range shown here is also not explained.

The red marked region is now defined in the figure caption and units are shown in the figure.

Table 1: I must admit that I failed to comprehend what is shown here, and the explanation 158-160 didn't help me really. What is the measured quantity, what is calculated from it?

Table 1 shows the chemical characterisation of the selected substances (e.g. listed in table 1 -left side): this includes, for example, hydrolysable carbohydrates and eHS, both expressed in mg per mg of substance, while Fe binding ligands are expressed in nM per mg of substance. Most of these were derived from existing papers (listed) and measured in inorganic synthetic seawater as stated in the text (we have added a short section to this effect to the methods). These numbers are used to derived LCarb and LEPS estimates and calculate their potential contribution to in situ ligands, as briefly outlined in the methods. We hope that the text added to the methods related to the humic materials considered and the data analysis has made this clearer.

Minor comments & typos

Line 59: "contrastingregions" -> contrasting regions *Done.*

Line 129ff: The sentence starting with "Whereas" is incomplete, I suspect some part has been lost in a previous copyediting operation. There is also no whitespace between this and the preceeding sentence. *This has been corrected.*

Line 380: "measure" -> measured. *Done*

Line 474: "similar .. than" does not sound right. *This section has been substantially modified to improve clarity.*

Line 474ff: The sentence starting with "Because" is missing a verb, I believe *This section has been thoroughly modified.*

Line 477: "apotential" -> a potential. *Done.*

POINT-BY_POINT REPLY TO REVIEWERS

We are very grateful for this second round of reviewers' comments, which have improved our manuscript. Our revised manuscript addresses all the comments and concerns raised. To show how the reviewers' comments have been addressed in our revised version of the manuscript, we copied their original reviews and added our reply below each comment in italics, with references to line numbers of the revised manuscript. Changes can also easily be assessed in the version with track changes shown and corresponding lines indicated in our reply. The indicated lines correspond to the manuscript with track changes shown throughout our reply.

REVIEWER COMMENTS

Reviewer #1 (Remarks to the Author):

Most of the reviewer comments have been addressed appropriately and the manuscript is far more inclusive and transparent than the previous version. There are some issues with figures, particularly fig 2, noted below. Some parts of the new manuscript do feel rushed with typos or referencing mistakes which are highlighted below, or text that assumes readers already have a greater understanding of ligands, humics etc than I would expect the general audience of this journal to be expected to have, and so it does still require some minor attention. There is still some reliance on correlations which was noted by multiple reviewers, although the text has been modified to tone down the language and a suitable rebuttal given.

We have modified Figure 2 as per reviewer's suggestions, checked for typos and appropriate reference inclusion. We have also added more information on ligands to make the text more accessible to the general audience .

There is however still an issue with the way the novelty of this study is presented. For example, in lines 188-190 at the start of the Discussion – it's great that there is now more data to further understand Fe biogeochemistry in the Southern Ocean, and this data deserves publication for that alone, as well as the expansion to including Carb and EPS, but as was pointed out by two different reviewers, the differing role of eHS in the Southern Ocean compared to elsewhere was already noted in Whitby et al. 2020 (see comments from R1 and R3). Furthermore, the difference in eHS contribution to FeL between the South Pacific, Southern Ocean and elsewhere was also recently highlighted in Whitby et al., 2024. It's fine to keep the text as it is, but it would be appropriate to reference the earlier studies here to demonstrate they are being built on and this new data augments and concurs with those earlier findings. The novelty is in the expansion of this complicated story with the inclusion of additional complementary measurements, and this still needs to be better highlighted.

We were already referencing Whitby et al. (2020) extensively throughout the text and we now include Whitby et al. 2024, but as per the reviewer suggestion, we kept the text as is. We modified the text in the abstract (l. 25-28) and other sections (l. 76-77, l, 219-220, l 325-328) to convey more clearly the novelty of this study.

Detailed comments:

The abstract is much improved, but it could still be clearer. Essentially, I think you want to say that you find that humic substances are an important part of the ligand pool for Fe in the Southern Ocean as has been found elsewhere, but that it is mostly produced in situ and composed of EPS, in contrast to other regions where terrestrial-derived humics may play a greater role(?). As written this doesn't really come across.

The text has been changed as per the suggestion and now reads "Here, we use a comprehensive dataset of electroactive humics and iron-binding ligands in contrasting regions across the Southern Ocean to show that humic substances are an important part of the iron binding ligand pool, as has been found elsewhere. However, we demonstrate that humics are mostly produced in situ and composed of exopolymeric substances from phytoplankton and bacteria, in contrast to other regions where terrestrially-derived humics are suggested to play a major role."

Figure 1B – the arrow is hiding the label of eHS, move the label so that it can be clearly seen

The figure has been modified and L is replaced by Fe-L throughout the text and figure to ensure consistency.

Figure 2:

- caption – is DDFe a typo? Also, what about FeL in this description. *Done*
- missing references within the caption text (just says 'refs' in multiple places). *The references have been checked.*
- The data for the Southwest Pacific is missing from the Fig 2a, or the description for 2b should clearly state that this figure also includes data for the Southwest Pacific as well as the Southern Ocean, otherwise this is misleading. *The panel b only shows SO data and the South Pacific is included in Fig. 2a.*
- I cannot see lines on the figures for 'Calculated DFe saturation of eHS is shown (dotted lines) considering HA (top) and FA (bottom)' on the pdf although they are weakly visible in the tracked changes document. *The lines have been thickened to be visible in the pdf.*
- The caption states 'Relationship between eHS and L is shown for (c) different ocean basins...' the axis labels instead show DFe vs L, not eHS. Either axes or caption incorrect. Again, d says 'for Southern Ocean' when actually it is Southern Ocean + Southwest Pacific. *The caption has been fully revised and panel b and d now only show SO data. As per Reviewer 3 suggestion, we also show the number*

of observations associated to our study as well as the total number of observations represented.

- L517 – typo in ligands. *Corrected.*

- For the Mediterranean Sea, only one reference is provided to Dulaquais et al. 2018, however that paper uses FeL data first published by Gerringa et al. 2017. If the Gerringa data is used it should be correctly cited. There is no Northeast Pacific FeL data in ref 14 and no other reference is provided so I'm not sure what data figure 2c and d is using for the Northeast Pacific. *The Gerringa reference has been added, and ocean basin have been simplified so the NE and NW distinction is not relevant anymore.*

- Is it necessary to distinguish between North Atlantic and Northwest Atlantic in the figure? In fact, shouldn't it be Northeast Atlantic, not Northwest, if it's referring to data from Dulaquais et al 2018? These confusions could be resolved by making it clearer to which references the data on the figure refer to. One solution would be to include the numbered references next to the basin name within the legend on each subplot, because it's difficult to associate the data on the figure with the correct references in the caption. *We have clarified accordingly by including references next to the corresponding ocean basin and by simplifying those so that for example only North Atlantic is shown without distinction of NE or NW.*

Figure 3 – 'as in ref 6' – ref 6 is Boyd et al., 2010? I don't think this is the correct reference. I cannot see the dotted line that represents the 100% contribution to FeL on the pdf (although I can see it on the tracked changes document). You may wish to just list the ocean basins along the y axis instead of numbers 1 – 6 with numbered references, it would be easier to follow. Also why does the x axis on D not start at 0? Finally, similar issues to Fig 2 for references.

We agree, reference 6 was from the original version of the manuscript and has now been corrected to 14 (Whitby et al. 2020). The 100% dotted lines were thickened to be visible in the pdf document. We kept ocean basin listed in the figure and caption as listing them instead of the 1-6 axis label would either reduce overall figure size and impair its reading or use precious journal space.

Figure 4 – I don't fully understand this figure, the caption could better explain what is being shown, what is meant by 'role on in situ Fe biogeochemistry'. What is 'Maximal Fe-L'?

Maximal Fe-L was replaced by Fe-L as it is the Fe-L calculated considering the different humics material and carb considered in the study. Data were compared to the measured in situ Fe-L to explore contribution to Fe-L and in-situ

complexing capacity. It is this latter comparison that allows us to infer their potential role on Fe biogeochemistry. The figure caption was extended to clarify the point made by this figure (l.613-618).

Specific comments:

All the specific comments below were addressed.

L35 – thee to the. *Done.*

L70 – ‘despite recent data suggest’ is missing something, e.g. ‘which suggests’ or ‘suggesting’ *Done “suggest that”*

L74 –what is the ‘Uniqueness of iron..’? *Done. “of iron binding organic...”*

L76 – move comma after life. *Done.*

L82-88 – give some more information on binding capacities, what it means to be similar to Suwannee river HS etc. The average reader without knowledge of humics will not understand this section.

The text has been modified and now reads (l.96-102) “Overall, the intercept was 1.3 nmol L⁻¹ and the slope was 0.045 nmol μg⁻¹ eHS (Supplementary Fig. 4). Thus, on average, 1 mg eHS could be associated with 45 nmol L⁻¹ Fe-L, which is close to the terrestrial SRHA value (32 nmol Fe mg⁻¹ SRHA)^{14,33,34}. Fe-L and eHS values associated with sedimentary input close to the Balleny Islands^{13,35} were amongst the highest measured and could explain a Fe binding capacity for eHS close to that of terrestrial humics.”

The close overall average DFe/eHS (Fig. 3b) discussed in the following paragraph is used to stress the relevance of SRFA and SRHA to assess Fe biogeochemistry at a global scale, yet probably not for the SO. The text reads (l.119-125) “ Surprisingly, similar average DFe:eHS were found across regions globally (18.4 ± 19.1 nmol Fe mg⁻¹ eHS, Fig. 3b); a value that is representative of SRFA and SRHA maximal Fe binding capacity or stoichiometry^{14,33}. This finding suggests that the role of humics in the global ocean could be assessed using Suwanee River reference material, although this approach might not be suitable for investigating the southernmost waters. For the SO, a median DFe:eHS of 8 nmol Fe mg⁻¹ eHS was observed, suggesting an undersaturation of eHS and/or Fe-L properties or sources different from SRHA and SRFA. “

L92 – is DFe supposed to be here or a typo. Yes, *it was removed.*

L89-92 – It could just show that humics are less saturated with DFe, which is not surprising in an Fe-limited ocean.

Yes, this is now stated in this paragraph, which has also been modified to make it more accessible to the wider audience as per the suggestion below (l. 109-111).

L89-100 – overall this section would be challenging for many readers to follow if they are not already experts in ligands and humics. I suggest making this a little more accessible to non-experts.

The text has been modified to “Although this finding supports the postulated role of eHS in Fe biogeochemistry across the global ocean^{14,19-23,36}, no correlation between DFe and eHS was observed for the SO (Fig. 2b) in contrast to all other oceanic regions for which comparable data are available (Fig. 2a, Supplementary Fig. 3, except for the South Pacific Ocean). In the global ocean, typically >99 % of the DFe is associated with Fe-L⁶; therefore, if eHS represents the bulk of Fe-L, one would expect a correlation between DFe and eHS unless (i) eHS are undersaturated with Fe, as might be anticipated in an Fe-limited region such as the SO, and/or (ii) a strong in situ competition between different metals and cations to bind the same eHS molecule exists^{6,13,14}. Similar eHS concentrations have been reported for several ocean basins and the Mediterranean Sea (Fig. 3a). The Arctic Ocean emerges as a notable exception, where eHS concentrations are elevated, likely due to strong riverine inputs and high concentrations of terrestrial humics in the transpolar drift^{20,21}. The DFe levels associated with eHS are within or smaller than the range observed for saturated Suwannee River FA and HA for a significant portion of the data collected in the North Pacific, North Atlantic and South Pacific Oceans. In the other regions, even lower values were measured (Fig. 3b). Together, competition between other metals and cations to bind eHS, and low DFe resulting in undersaturated eHS, will typically lower the measured DFe:eHS values. Surprisingly, similar average DFe:eHS were found across regions globally (18.4 ± 19.1 nmol Fe mg⁻¹ eHS, Fig. 3b); a value that is representative of SRFA and SRHA maximal Fe binding capacity or stoichiometry^{14,33}. This finding suggests that the role of humics in the global ocean could be assessed using Suwannee River reference material, although this approach might not be suitable for investigating the southernmost waters. For the SO, a median DFe:eHS of 8 nmol Fe mg⁻¹ eHS was observed, suggesting an undersaturation of eHS and/or Fe-L properties or sources different from SRHA and SRFA. Indeed, the relationships between DFe, eHS, and Fe-L indicate a distinct nature of organic matter – Fe interactions in this southernmost ocean (Fig. 2, Supplementary Figs. 3 and 5). “

“

L108 – “first served”. The text has been changed accordingly.
L116 – what is meant by “the bulk of eHS... cannot be described by standard FA and HA substances” – are you inferring that these eHS have a different binding capacity due to being marine-derived? Or that other ligands are important in these regions?

The following was added “For the SO, a median DFe:eHS of 8 nmol Fe mg⁻¹ eHS was observed, suggesting an undersaturation of eHS and/or Fe-L properties or sources different from SRHA and SRFA.”

L121-122 – I do not understand this sentence (weaker relations to what?), plus it has multiple typos - “Coexistence of numerous compounds [change to compounds] typically results in “weaker” relations that [change to than] expected on known processes²⁸”

This sentence (l.149) has been changed to “Coexisting nutrients and organic matter sources and degradations processes often weaken the observed relationships between

the biomass of phytoplankton and bacteria with Fe-L compared to what is expected based on our understanding of known processes at play³⁰

L124 – ‘to specific microorganisms’ – I did not see any data presented here on comparing to individual microorganisms, only to bacteria vs phytoplankton. Rephrase this for clarity. *Yes, this was replaced by phytoplankton and bacteria.*

L140 – add here “EPS, which compose part of the eHS pool, ...” (or however you wish to associate them) for clarity. *Done.*

L174 – add space before ‘We’. *Done.*

L240 – add stop after inputs. *Done.*

L268 – the lack of ‘a’ marine humic standard. *Done.*

L277 – add space between ‘they modulate’. *Done.*

L288 – should this be particles. *Yes, the text has been changed to “as important particles for carbon export.”*

Some comments from previous review round have no response, e.g. R1 asks ‘only 30s purge?’ with no response, and ‘A single ligand class was obtained’ doesn’t really address the question as to why only one ligand class was obtained when usually SA gives 2. Was an attempt made to fit multiple and it was not possible? Why might this be the case as this is unusual. I don’t consider receiving responses to these queries should be an impediment to publication, but if further review iterations occur, it would be good to see the explanation.

The purge was only 30 sec as compressed air was used (and not N₂) and short purge time is thus appropriate as no change in the solution to be analyzed is made. This approach was used by several laboratories and was published in several journals.

Please find below representative results for multiple ligands output using ProMCC fit to demonstrate that several ligands were not appropriate here. Please consider the low concentration for L2 and log K (expressed as per Fe’) values but amongst all errors equaling 100 % for this second class of ligands and log K.

Station 7 40m													
Fit	Slope	M(amb)	M(ln/Free)	+/-	L1	+/-	logK1	+/-	L2	+/-	logK2	+/-	AVG error
	1.90E+05	2.30E-10	4.43E-12	2.73E-12	1.13E-09	6.00E-11	10.75	64	5.32E-10	5.19E-10	8.48	8.48	5827
Data=	Conc.	Intensity	Use										
1	2.30E-10	3.24E-05	0										
2	2.30E-10	4.96E-05	0										
3	6.30E-10	4.76E-05	1										
4	6.30E-10	3.94E-05	1										
5	1.03E-09	7.05E-05	1										
6	1.03E-09	7.63E-05	1										
7	1.23E-09	1.01E-04	1										
8	1.23E-09	9.84E-05	1										
9	1.43E-09	1.35E-04	1										
10	1.43E-09	1.48E-04	0										
11	1.63E-09	1.51E-04	1										
12	1.63E-09	1.54E-04	1										
13	1.83E-09	1.81E-04	1										
14	1.83E-09	1.82E-04	1										
15	2.23E-09	2.37E-04	1										
16	2.23E-09	2.45E-04	1										
17	2.63E-09	3.13E-04	1										
18	2.63E-09	3.08E-04	1										
19	3.03E-09	3.81E-04	1										
20	3.03E-09	3.99E-04	0										
21	3.63E-09	5.22E-04	0										
22	3.63E-09	4.76E-04	1										
23	4.23E-09	5.96E-04	1										
24	4.23E-09	5.73E-04	1										
25	5.23E-09	7.47E-04	1										
26	5.23E-09	7.84E-04	0										
Station 7 200m													
Fit	Slope	M(amb)	M(ln/Free)	+/-	L1	+/-	logK1	+/-	L2	+/-	logK2	+/-	AVG error
	1455533.11	2.10E-10	1.96E-12	2.05E-12	8.49E-10	1.66E-10	11.22	0.28	2.34E-18	2.28E-18	8.36	8.36	17758
Data=	Conc.	Intensity	Use										
1	2.10E-10	5.83E-05	1										
2	2.10E-10	7.32E-05	1										
3	6.10E-10	1.66E-04	1										
4	6.10E-10	1.93E-04	1										
5	1.01E-09	4.73E-04	1										
6	1.01E-09	5.02E-04	1										
7	1.21E-09	7.31E-04	1										
8	1.21E-09	7.00E-04	1										
9	1.41E-09	9.92E-04	1										
10	1.41E-09	9.53E-04	1										
11	1.61E-09	1.35E-03	1										
12	1.61E-09	1.27E-03	1										
13	1.81E-09	1.71E-03	1										
14	1.81E-09	1.65E-03	0										
15	2.21E-09	1.71E-03	0										
16	2.21E-09	1.68E-03	0										
17	2.61E-09	2.88E-03	1										
18	2.61E-09	2.77E-03	1										
19	3.01E-09	3.54E-03	1										
20	3.01E-09	3.63E-03	0										
21	3.61E-09	4.44E-03	1										
22	3.61E-09	4.25E-03	1										
23	4.21E-09	4.66E-03	0										
24	4.21E-09	4.45E-03	0										
25	5.21E-09	6.48E-03	0										
26	5.21E-09	6.28E-03	0										

Reviewer #2 (Remarks to the Author):

After reviewing the revised manuscript, I am pleased to confirm that the authors have significantly improved the paper. Many concerns raised in earlier reviews, including mine, have been addressed thoughtfully, resulting in a more cohesive and compelling narrative.

The flow of the manuscript has greatly improved, and even sections that I initially found problematic—such as the statistical analysis—now make more sense within the revised

structure of the text. By integrating various components effectively, the authors have summarized their findings in a way that better conveys the significance and novelty of the study.

The methodological clarifications and the inclusion of additional references have strengthened the paper's foundation. The authors successfully addressed concerns about previous misinterpretations of data and justified their analytical approaches with clearer explanations. The revised figures and improved statistical presentation enhance the paper's readability and reliability.

In conclusion, this paper presents a valuable and well-substantiated contribution to the field, and I recommend it for publication.

We thank this reviewer for their positive evaluation of our revised version.

Reviewer #3 (Remarks to the Author):

Review of the first revised version of "Marine biogenic humic substances control iron biogeochemistry across the Southern Ocean" by Hassler et al, submitted to Nature Communications

In their revised version, the authors have corrected my main concern with the first version of the paper, which was the misrepresentation of the conclusions drawn in the study by Whitby et al., 2020. They have also clarified many of their unclear statements and now discuss much better how the different, often operationally defined, subclasses of marine DOM (humic substances, electroactive humics, ligands, EPS) relate to each other. Although the methods used do not allow to directly quantify the role of EPS in binding iron, the authors now quite convincingly show that EPS has the possibility to be an important contributor to Fe-binding ligands in the Southern Ocean and South Pacific. I would therefore now recommend publication of the study in Nature communications, after some minor revisions.

We thank this reviewer for their positive evaluation of our revised version.

Minor comments

Line 50: 'Iron bioavailability varies ...!'; the authors of course know very well that the bioavailability, as it is defined in citations 12 and 13 is not a property of seawater iron chemistry alone, but is also influenced by the uptake mechanisms of the phytoplankton species involved; maybe it would be helpful for the reader if this was mentioned briefly in this paragraph.

The text has been changed accordingly with the following addition “: Iron bioavailability is not controlled by seawater chemistry only, but it is also influenced by the biological uptake mechanisms at play, as well as biological competition for Fe acquisition^{9,11-13}.”

Line 82 ff: I find the statement made here (especially 'suggest' and 'most') maybe a bit too strong for the fact that a) the intercept 1.3 nmol/L ligand for zero eHS is not negligible, and b) that the calculation has been done using the lab-based maximum Fe binding capacity for Suwanee river humics, i.e. neglecting the competition of other cations for binding.

We replaced “most” by “a substantial fraction” to tune down this statement. We removed the “rather low” associated with the intercept. The excess of ligands could represent a possible explanation for this intercept yet adding it in the text will most likely confuse the general reader and it will break the flow of the point made in this paragraph. Hence this was not added. The text reads (l. 96-102) “Overall, the intercept was 1.3 nmol L⁻¹ and the slope was 0.045 nmol μg⁻¹ eHS (Supplementary Fig. 4). Thus, on average, 1 mg eHS could be associated with 45 nmol L⁻¹ Fe-L, which is close to the terrestrial SRHA value (32 nmol Fe mg⁻¹ SRHA)^{14,33,34}. Fe-L and eHS values associated with sedimentary input close to the Balleny Islands^{13,35} were amongst the highest measured and could explain a Fe binding capacity for eHS close to that of terrestrial humics.” We rather focused on the fact that high Fe-L and eHS values reported close to Balleny Island strongly affect the correlation observed (slope and intercept). This last addition is also relevant to the fact that this general trend from linear correlation does not match the median 8 nmol DFe per mg eHS shown in Fig 3b.

We further added the competition for cations binding to the same eHS molecule to explain the absence of relationship between DFe and eHS (l. 107-111) “ In the global ocean, typically >99 % of the DFe is associated with Fe-L⁶; therefore, if eHS represents the bulk of Fe-L, one would expect a correlation between DFe and eHS unless (i) eHS are undersaturated with Fe, as might be anticipated in an Fe-limited region such as the SO, and/or (ii) a strong in situ competition between different metals and cations to bind the same eHS molecule exists^{6,13,14}. “ and equal to lower value of DFe/eHS observed in Fig. 3b than SRFA and SRHA Fe binding capacity by adding the following (l.117-119): “Together, competition between other metals and cations to bind eHS, and low DFe

resulting in undersaturated eHS, will typically lower the measured DFe:eHS values.”

Besides, I do not understand to which graph the mentioned intercept and slope in line 85 belong, to Figure 2d? Maybe it would help here whether the nmol/L are actually nmol/L of Fe or of 'Fe-L'?

It is supplementary Fig. 4 which is now mentioned in the text.

And, to add to my confusion: If the slope is 0.045 nmol/L / (ug eHS) (which, I assume should be 0.045 nmol/L / (ug eHS / L)), why is then 1 ug eHS equivalent to the inverse $1/0.045 = 22.2$? This whole calculation is unclear.

Yes, and we thank the reviewer for pointing this out. The slope is 0.045 nmol/L Fe/ (ug eHS / L), hence it is 45 nmol Fe/mg eHS. The text has been amended accordingly (see above).

Line 134 ff, '...Fe bioavailability, more than Fe concentration, is key to phytoplankton photosynthesis and growth': I have some difficulty with this statement. The positive correlations of Fv/Fm with Fe-L and eHS indeed suggest that iron that is bound to ligands and electroactive humics is bioavailable. But then why is there no correlation observed to DFe? Does this suggest that some part of the DFe is NOT bioavailable, and which part would that be, given that ligand concentrations shown in Fig 2d are fairly high, and ligands are mostly unsaturated with Fe? Colloidal DFe? Iron bound to some strong ligands that are not detected?

This statement has been changed and colloidal consideration is listed yet we did not go too much in this direction as this relies on rather weak R and colloidal Fe data is not available here; colloidal Fe is not at the center of our story. The text now reads (l. 166-175) “Despite the rather low R values, these correlations suggest a ligand-mediated Fe bioavailability to phytoplankton that cannot be predicted from DFe alone, as previously observed for natural phytoplankton assemblages⁹. The discrepancy between DFe concentrations and ligand-bound Fe bioavailability could be related to the occurrence of a significant proportion of DFe that is less bioavailable because it is associated with colloids. Colloidal Fe plays a critical role in Fe residence time, with consequences for the dynamics of marine ecosystems³¹; unfortunately, we have no measurements of colloidal Fe and thus cannot explore whether these forms are responsible for the lack of correlation between DFe and the biomass of phytoplankton or bacteria.”

It could also be due to undetected strong ligands as suggested, but as we were able to titrate and detect DFB in our quality control test in the lab (see Cabanes et al. 2020), we

would therefore prefer not to extend too much on this as it is unclear how much of the strong ligands are not detected and this is also out of the scope of this study.

Line 223: Given that it is based to a large part on assumed iron-binding capacities for humics, the statement 'Our study identifies EPS as the bulk of Fe-binding marine humics and ligands' is maybe a bit strong.

The text has been changed to (l. 271-273) "Our study suggests that EPS represent the bulk of Fe-binding marine humics and ligands, and are a key component for Fe biogeochemistry in the upper 1000 m of the SO water column."

Line 227: The citation 56, given as support for the association of hydrothermally derived iron with (e)HS, is rather indirect: It deals with humics in hydrothermally influenced sediments. Is there more direct evidence for the association of humics with Fe in hydrothermal plumes?

*This citation has been replaced by Yang, L., Zhuang, W.-E., Chen, C.-T., A., Wang, B.-J. & Kuo, F.-W. Unveiling the transformation and bioavailability of dissolved organic matter in contrasting hydrothermal vents using fluorescence EEM-PARAFAC. *Wat. Res.* **111**, 195-203 (2017). Four hydrothermal vents fluids were analysed. Strong humics components are shown from high humification index, C1 PARAFAC component, as well as low bacterial degradability.*

Figure 1B: would it be possible to separate the labels eHS and L in the figure somewhat? And why is 'Fe-L' called 'L' in the figure and caption? In the caption it is mentioned that logK and Carb are not statistically represented. Does that mean that only a low part of their variance is explained by the first two EOFs? Then I would remove them from the plot all together, right?

As per suggestion, Fe-L was homogenized throughout the text, including in Fig. 1b. Even if log K and Carb are not strictly statistically represented, we think it is interesting to show how they related to other data. Yes, only 22% and 53% of log K and Carb, respectively are described by the 2 EOFs as stated in the caption. Removing them does not change PCA as per the figure below. We thus maintained these 2 parameters in this figure. The only difference is in the explanatory percentage from the 2 EOFs that goes from 64.3 % considering all dataset and up to 86 % considering only the significant dataset, yet it does not change how eHS and L relate, which is the central point made by this figure.

Figure 2: The dotted lines are not visible in my printout; they are visible in the pdf at some magnification. Maybe make them somewhat more prominent? In the caption, line 512: 'DDFe' -> 'DFe', Maybe ligands should also be mentioned in the bold part of the caption? Obviously some references were intended to be included but weren't '(refs)'. What does the 'mostly this study' mean?

For the SO data we have now included number of observations from Whitby et al., 2020 and from our study, this also clarify how our data extend previous observations. The following text was added to Fig 2 caption “. For the SO, our study contributes to 106 out of the 118 observations on panel b, and all the data represented in panel d as no Fe-L data are available in ¹⁴” and to Fig. 3 “Ocean¹⁴ (mostly this study, n=106 for panels a and b, n= 70 for panels c and d).“

In Fig. 2, the line corresponding to SRFA and SRHA Fe binding capacity was made thicker to be visible in the pdf. The DDFe was corrected to DFe in the caption and the caption was fully revised to match figure and showing appropriate references.

Line 558 'following GEOTRACES guidelines': This merits a citation.

Citation to the Cookbook has been included.

Line 634: How large, approximately, is the error in the calculated complexation capacity, as calculated from the errors in L_insitu, DFe and the conditional stability constants?

Considering typical error propagation using average values and standard deviations (Fantner, G., 2011, A brief introduction to error analysis and propagation, https://www.epfl.ch/labs/lben/wp-content/uploads/2018/07/Error-Propagation_2013.pdf):

$$\text{For } eL = L - DFe \quad SD\ eL = \text{square root } ((SD\ L)^2 + (SD\ DFe)^2)$$

*For binding capacity (referred as BC below) $eL * K$*

$$(SDBC / BC) = \text{square root}((SD\ eL/eL)^2 + (SDK/K)^2)$$

Here we considered the average value and SD (% , n=70) : $L = (2.76 \pm 2.75) * 10^{-9} M$, $D_{Fe} = (0.177 \pm 0.135) * 10^{-9} M$, $K = 10^{11.28} \pm 10^{9.66}$ (2.4% on average), average $eL = 2.6 * 10^{-9} M$ and average BC for Fe-L = 881.6.

The SD for eL is thus $2.75 * 10^{-9} M$; and the SD for BC is 932.6.

Please note that the 5th and 95th percentiles shown on Fig. 4 also indicate data range and that actual standard deviation is already shown in the figure considering individual calculation for each sample rather than average as done above. Thus no change was made to the text.

Typos and small comments

All typos and small comments have been addressed, except for one (please see our reply below).

Line 35: thee -> the

Line 48 and several following: I find it somewhat unfortunate to abbreviate iron-binding ligand concentrations as Fe-L, as this might be mistaken for the concentration of iron bound to ligands. Maybe LFe (possibly with Fe as subscript) is less confusing and more in line with other papers on the subject?

We homogenize the use of Fe-L throughout the text and figures. We decided not to use LFe as the text is referring of calculated Fe binding ligands using different humics as LFA, LHA and so on. Hence this notation only refer to calculated value used to estimate potential contribution and not to measure values.

Line 92: I think 'South Pacific)DFe' should just be 'South Pacific'.

Line 121: 'that' -> 'than', 'on' -> 'from'

Line 138: Here ligands are suddenly abbreviated as 'L', not as 'Fe-L'. Why?

Line 189: 'humic' -> 'humics'

Line 517: 'liagnds' -> 'ligands'. There should be a '1,' before 'Arctic'.

Line 650: '_H. Forrer' -> ' H. Forrer'